# NOTCH1 Signaling in Head and Neck Squamous Cell Carcinoma

**DOI:** 10.3390/cells9122677

**Published:** 2020-12-12

**Authors:** Pooja A. Shah, Chenfei Huang, Qiuli Li, Sawad A. Kazi, Lauren A. Byers, Jing Wang, Faye M. Johnson, Mitchell J. Frederick

**Affiliations:** 1Department of Thoracic/Head & Neck Medical Oncology, The University of Texas MD Anderson Cancer Center, Houston, TX 77030, USA; pshah3@mdanderson.org (P.A.S.); lbyers@mdanderson.org (L.A.B.); 2Bobby R. Alford Department of Otolaryngology, Baylor College of Medicine, Houston, TX 77030, USA; chuangfeifei@gmail.com (C.H.); Mitchell.Frederick@bcm.edu (M.J.F.); 3Department of Head and Neck Surgery, Sun Yat-sen University Cancer Center, State Key Laboratory of Oncology in South China, Collaborative Innovation Center for Cancer Medicine, Guangzhou 510060, China; liql@sysucc.org.cn; 4School of Natural Sciences, University of Texas, Austin, TX 78712, USA; Sawad.Kazi@utexas.edu; 5The University of Texas Graduate School of Biomedical Sciences, Houston, TX 77030, USA; jingwang@mdanderson.org; 6Department of Bioinformatics and Computational Biology, The University of Texas MD Anderson Cancer Center, Houston, TX 77030, USA

**Keywords:** head and neck squamous cell carcinoma, NOTCH1, tumor suppressor, mutation, synthetic lethal, phosphatidylinositol 3-kinase

## Abstract

Biomarker-driven targeted therapies are lacking for head and neck squamous cell carcinoma (HNSCC), which is common and lethal. Efforts to develop such therapies are hindered by a genomic landscape dominated by the loss of tumor suppressor function, including *NOTCH1* that is frequently mutated in HNSCC. Clearer understanding of NOTCH1 signaling in HNSCCs is crucial to clinically targeting this pathway. Structural characterization of *NOTCH1* mutations in HNSCC demonstrates that most are predicted to cause loss of function, in agreement with NOTCH1’s role as a tumor suppressor in this cancer. Experimental manipulation of NOTCH1 signaling in HNSCC cell lines harboring either mutant or wild-type *NOTCH1* further supports a tumor suppressor function. Additionally, the loss of NOTCH1 signaling can drive HNSCC tumorigenesis and clinical aggressiveness. Our recent data suggest that NOTCH1 controls genes involved in early differentiation that could have different phenotypic consequences depending on the cancer’s genetic background, including acquisition of pseudo-stem cell-like properties. The presence of *NOTCH1* mutations may predict response to treatment with an immune checkpoint or phosphatidylinositol 3-kinase inhibitors. The latter is being tested in a clinical trial, and if validated, it may lead to the development of the first biomarker-driven targeted therapy for HNSCC.

## 1. Introduction 

Head and neck squamous cell carcinomas (HNSCCs) arise mainly from mucosal surfaces of the oral cavity, oropharynx, pharynx, larynx, and sinonasal cavity. Globally, in 2018, HNSCC accounted for 890,000 new cancer cases, making it the seventh most common cancer, and 450,000 deaths [1]. Alcohol and tobacco use and infection with high-risk types of human papillomavirus (HPV) are the main risk factors for HNSCC. Standard therapeutic options for locally advanced HNSCC consist of surgery, irradiation, chemotherapy, and combinations of them. Nevertheless, the overall five-year survival rate in treated HNSCC patients remains about 50% [2], highlighting the urgent need for better treatment options. Current immunotherapeutic approaches for recurrent or metastatic HNSCC are associated with modest response rates of 14–18%, six-month progression-free survival rates of 23%, and a one-year survival rate of 36% [3,4,5,6,7]. Although immunotherapy has a striking effect in some HNSCC patients, the majority still experience progression [3,4,5,6,7,8,9] emphasizing the urgent need for improved therapeutics.

Efforts to develop more targeted therapies for HNSCC are hampered by the genomic landscape of the disease, which is dominated by the loss or inactivation of tumor suppressor genes, and very few directly druggable oncogenic drivers [10,11,12,13]. A promising alternative strategy would be to identify and target co-dependencies that arise in tumors owing to the loss of specific tumor suppressors, in a synthetic lethal manner. *NOTCH1* is one of the most frequently mutated tumor suppressor genes in HNSCC [14]. We recently reported that HNSCC cell lines harboring *NOTCH1* loss of function (LOF) mutations are highly dependent on phosphatidylinositol 3-kinase (PI3K) signaling and that treatment with drugs blocking this pathway leads to tumor cell death in this genomic subtype [15]. However, other groups have reported data suggesting that *NOTCH1* has oncogenic properties in a subset of HNSCC cases [16,17], leading to the hypothesis that the NOTCH1 pathway may have dual function in this tumor type. A clearer understanding of NOTCH1 signaling and its role in HNSCC may be key to clinically targeting this pathway in patients. Given the importance of *NOTCH1* in HNSCC, this review will provide an overview of NOTCH1 biology and signaling in HNSCC, a comprehensive examination of *NOTCH1* mutations in HNSCC and other squamous cancers, and a discussion of NOTCH1’s potential dual function in HNSCC. Additionally, this review will provide an overview of clinical correlations and present our recent approach for targeting HNSCC patients whose tumors harbor *NOTCH1* mutations.

## 2. *NOTCH1* Encodes for a Conserved Receptor Regulating Transcription and Cell Fate

*NOTCH1* belongs to the NOTCH family of receptors (*NOTCH1-4*), which is part of an evolutionarily conserved signaling pathway innate to all multicellular organisms and plays a crucial role in embryogenesis and tissue homeostasis. NOTCH receptors regulate essential cellular functions linked with cell fate specification, including proliferation, differentiation, apoptosis, and stem cell maintenance [18,19]. NOTCH1 is the most extensively studied and characterized NOTCH family member because its mutation prevalence among human cancers is higher than that for other members [20,21]. Aberrant NOTCH1 signaling is implicated in the progression of various cancer types including breast cancer, leukemias, HNSCC and squamous cancers of the skin, esophagus, cervix, and lung [10,11,22,23,24,25,26]. NOTCH1 can function as either a tumor promoter or suppressor largely depending on the cellular context [27].

NOTCH1 receptor signaling is more easily understood in the context of the family’s protein structure, which is illustrated for NOTCH1 and its ligands in Figure 1. NOTCH1 is first synthesized as a precursor protein that folds and initially undergoes cleavage to produce extracellular and intracellular/transmembrane-associated peptides that remain tightly bound through their hetero-dimerization domains (HDs) during transport to the cell plasma membrane. NOTCH1 signaling is initiated in a juxtacrine fashion through binding to one of five canonical ligands (Jagged-1 (JAG1), JAG2, Delta-like ligand 1 (DLL1), DLL3, and DLL4) expressed on the surface of neighboring cells. These ligands interact with extracellular epidermal growth factor (EGF)-like repeat binding domains spanning extracellular NOTCH1, causing conformational changes that open up NOTCH1 to cleavage by a disintegrin and metalloproteinase (ADAM) and subsequently the γ-secretase complex [28,29,30]. Cleavage occurs within the transmembrane (TM) region, releasing the intracellular NOTCH1 domain (ICN1), sometimes referred to as NOTCH intracellular domain (NICD) or cleaved (cl) NOTCH1, which migrates to the nucleus where it regulates transcription of many genes.

In the nucleus, ICN1 forms a ternary complex with recombination signal binding protein Jκ (RBP-Jκ, also called Epstein–Barr virus (EBV) C promoter binding factor 1 (CBF1)/Suppressor of Hairless (Su (H))/Lag-1 (CSL)) and mastermind-like transcriptional coactivator 1 (MAML1). The ICN1/RBP-Jκ/MAML1 complex activates the transcription of canonical downstream NOTCH1 targets from the Hairy and enhancer of split-1 (*HES)* and Hairy/enhancer-of-split related with YRPW motif protein 1 *(HEY*) gene families [17], which are themselves transcriptional repressors. Additionally, activated ICN1/RBP-Jκ/MAML1 recruits multiple other coactivators or repressors of gene expression, including histone modifiers [31]. NOTCH1 contains an RBP-Jκ-associated molecule (RAM) domain, seven ankyrin (ANK) repeat regions, a transcription activation domain (TAD) and proline-, glutamate-, serine-, and threonine-rich (PEST) domain. The difference between NOTCH family receptors lies in the number of EGF repeats and the presence or absence of a TAD (Figure 1). The C-terminal PEST domain in NOTCH is a target for binding F-box proteins (e.g., FBXW7), components of a ubiquitin protein ligase complex that degrades activated ICN to temporally limit signaling. Certain leukemias arise with inactivating *FBXW7* mutations in the cancer cells that complement NOTCH1 activation by preventing ICN1 degradation [32].

## 3. *NOTCH1* Mutations in HNSCC

### 3.1. Initial Identification of NOTCH1 Mutations

Advances in next generation sequencing over the past decade have accelerated the identification of cancer driver genes, particularly those with somatic mutations, in more than 30 tumor types [33]. HNSCC was one of the first cancers for which next generation sequencing was used to perform whole exome sequencing (WES) with a relatively large number of patient tumors. In a collaborative study by our group at the University of Texas MD Anderson Cancer Center and investigators at the Johns Hopkins University School of Medicine, WES performed with 32 primary tumors revealed frequent mutations of *TP53*, *NOTCH1*, *CDKN2A*, *PIK3CA*, *FBXW7*, and *HRAS* [10]. In that same work, focused sequencing of a validation cohort of 88 additional patient tumors established a *NOTCH1* mutation prevalence rate of 15% in HNSCC cases. An independent study led by researchers at the Broad Institute and the UPMC Hillman Cancer Center demonstrated a similar incidence of frequently mutated genes and *NOTCH1* mutations in 11% of HNSCC samples [11]. The patterns of *NOTCH1* mutations in these two studies were similar. Nonsense and additional types of truncating mutations (e.g., frameshift, splice site) that prematurely terminated the native NOTCH1 protein sequence before the C-terminal ANK repeats occurred more frequently than expected by chance, implying LOF mutations. Most missense mutations in both studies occurred within the EGF-like ligand binding domains, which was consistent with the hypothesis that these point mutations likely inactivate NOTCH1 function in tumors by interfering with receptor ligand binding.

The locations and patterns of *NOTCH1* mutations in HNSCC were in sharp contrast with the established *NOTCH1* activating mutations in hematopoietic malignancies, as the latter are localized in the HD and PEST domains. Moreover, loss of both wild type (*wt*) alleles at the *NOTCH1* locus occurred in roughly 40% of *NOTCH1* mutant HNSCC cases in the MD Anderson/Hopkins study. The investigators in the two early genomic studies described above were the first to propose a tumor-suppressive function of NOTCH1 in HNSCC as well as other human solid tumors. A tumor-suppressive role for NOTCH1 in human cancer was consistent with results of a much earlier study by Nicolas et al. [34] who reported that conditional knockout of Notch1 in mouse skin promoted cutaneous tumor formation. Additionally, inhibition of Notch signaling using dominant negative MAML1 expression in the epidermis led to cutaneous squamous cell carcinoma (SCC) in mice [35].

### 3.2. Confirmation of NOTCH1 Mutations in HPV-Negative and -Positive HNSCC

Following the initial discovery of *NOTCH1* mutations in HNSCC, a number of groups reported frequent *NOTCH1* mutations in multiple HNSCC patient cohorts spanning diverse geographical regions and etiological risk factors (Table 1). Researchers have found truncating mutations randomly distributed throughout *NOTCH1* and missense mutations mapping to the extracellular EGF-like ligand binding domains in cohorts from Asia [36,37], South Asia [38], Europe [39], Latin America [39], and the United States [10,11,13,40,41]. The same spectrum of *NOTCH1* mutations is apparent regardless of whether smoking and alcohol use or smokeless tobacco use is the predominant risk factor [36,37,39,40]. Among HPV-negative enriched HNSCC patient cohorts, the reported frequencies of *NOTCH1* truncating and missense mutations range from 4% to 10% and from 7% to 48%, respectively. The prevalence rate for *NOTCH1* mutations is 17% in white and other non-Chinese HPV-negative HNSCC patient cohorts but 40% in Chinese patients.

Genomic studies of *NOTCH1* mutations in HPV-driven oropharyngeal cancers (Table 1) demonstrated a spectrum of mutations similar to that in HPV-negative HNSCCs [10,13,40] that are presumably inactivating. The average frequency rate for *NOTCH1* mutations in HPV-positive oropharyngeal cancers is 10%, which is roughly half of that in HPV-negative HNSCC. Additional genetic evidence supporting a role for *NOTCH1* in carcinogenesis of HPV-driven squamous carcinomas was reported by Zhong et al. [44] who used an elegant model of transposon-mediated insertional mutagenesis (i.e., “Sleeping Beauty”) in mice with conditional co-expression of HPV E6/E7 in basal epithelium. In this model, mice were treated with chemical carcinogens to further promote tumors of the skin or oral cavity and *NOTCH1* was located in one of the most significant transposon common insertion sites for HPV-E6/E7-driven tumors. In the same study, the effect of *NOTCH1* LOF in the absence of transposon insertional mutagenesis was also examined in mice conditionally expressing mutated KRAS (KRAS^G12D^) or E6/E7 plus KRAS^G12D^ with hemizygous or homozygous *NOTCH1* LOF. Interestingly, tumor formation and growth rate were increased equivalently by the loss of either one or both *NOTCH1* alleles when E6/E7 was present with KRAS^G12D^; whereas loss of both *NOTCH1* alleles was required to promote tumor growth when E6/E7 was absent. Collectively, these data suggest that *NOTCH1* LOF can contribute to carcinogenesis of both HPV-positive and -negative HNSCC, but that haploinsufficiency of *NOTCH1* is sufficient when E6/E7 drivers are present too. This hypothesis may explain the detection of fewer somatic *NOTCH1* mutations in HPV-positive HNSCC cohorts, as there could be a reduced necessity to inactivate both alleles in this genomic subtype.

### 3.3. Structural Characterization of NOTCH1 Mutations

Because the locations and patterns of *NOTCH1* mutations in HNSCC and T cell acute lymphoblastic leukemia (T-ALL) were distinct, we pooled data available from genomic studies in the Catalogue Of Somatic Mutations In Cancer (COSMIC) database and The Cancer Genome Atlas (TCGA) Pan-Cancer Analysis of Whole Genomes Consortium in order to compare the structure of *NOTCH1* mutations in T-ALL and HNSCC patients. We filtered missense mutations for their potential impact on function using individual Protein Variation Effect Analyzer (PROVEAN) and Sorting Intolerant From Tolerant (SIFT) scores, which are measures of how damaging an amino acid variation is predicted to be, and only included them in the analysis if either or both scores were considered potentially damaging or deleterious. We identified a total of 350 and 1349 potentially impactful mutations (including missense, truncating, and insertion and deletion (INDEL) mutations) in HNSCC and T-ALL, respectively, and plotted them according to their locations (Figure 2). A difference in the spatial locations of truncating mutations between these two cancer types is highly apparent, with the vast majority of truncating mutations observed in HNSCC occurring before the TAD. In contrast, most of the truncating mutations in T-ALL cases occurred between the C-terminal PEST domain and about midway into the TAD. C-terminal truncating mutations that delete the PEST domain in T-ALL cases are known to prevent ICN1’s interaction with FBXW7 and contribute to prolonged nuclear signaling by ICN1 [25]. Because canonical NOTCH1 signaling is enhanced in T-ALL, it can be inferred from the observed distribution of mutations in Figure 2 that truncations happening before the approximate midpoint of the TAD likely weaken or inactivate ICN1 activity because they are practically nonexistent in T-ALL. Consequently, at least a portion of the TAD domain is likely required for full NOTCH1 function, consistent with reports that the TAD domain is required for induction of T-cell leukemia [45]. This is further supported by the distribution of truncations in HNSCC, where the vast majority occur before the TAD and are likely inactivating because they lack a crucial TAD region for activity. The approximate amino acid coordinate for this distinction or functional boundary in the TAD is drawn in Figure 2, so that truncation mutations to the left of the dotted line are highly likely to be inactivating and truncations found to the right are very likely to be activating.

A second hotspot for activating *NOTCH1* mutations in T-ALL patients is within the HD domain (Figure 3). Missense and INDEL mutations in this region are known to lead to ligand-independent NOTCH1 activation and signaling due to destabilization of the surrounding regions that negatively regulate protease cleavage and ordinarily prevent ligand-independent activation [46]. As expected, the proportion of missense and INDEL mutations observed for HNSCC in the HD domain is extremely low. Missense mutations in HNSCC appear distributed across most of the *NOTCH1* gene; however, their distribution is not completely random.

We performed a rudimentary analysis to compare the number of missense mutations in specific NOTCH1 protein domains with that expected by simple chance. Using the reference amino acid codons for NOTCH1, we counted the possible amino acid substitutions that could occur within each domain from every point mutation possible and filtered them according to their PROVEAN or SIFT scores that predicted an impactful change. We then calculated the probability of a potentially impactful mutation occurring in a domain by dividing the number of possible impactful amino acid substitutions in that domain by the total number of potentially impactful substitutions for the entire NOTCH1 protein. Next, we multiplied the domain-specific probability by the total number of observed potentially impactful missense/INDEL mutations observed in either HNSCC or T-ALL cases to calculate the expected number of mutations in each domain. Finally, we plotted the observed and expected number of impactful mutations (excluding nonsense and frameshift) in each domain for HNSCC and T-ALL to generate the mutation enrichment plots in Figure 3. Observed missense mutations are enriched in the EGF2, EGF4, EGF8, EGF9, EGF10, EGF11, EGF12, EGF29, EGF31, RAM, and ANK1 domains in HNSCC, suggesting that mutations in these regions are likely to inactivate NOTCH1 signaling. Missense mutations observed less frequently for the other domains in HNSCC tumors may represent passenger events. The repeat regions extending from EGF8 to EGF12 have been implicated in ligand binding based on experimental manipulation and X-ray crystallography [47], so mutations in these domains likely prevent or inhibit NOTCH1 signaling. In support of this conclusion, we previously demonstrated loss of NOTCH1 signaling in the established HNSCC cell line HN31 that carries a homozygous *NOTCH1* mutation (C478F) in the EGF12 domain [12]. The same amino acid is found mutated in two different HNSCC patient tumors from the TCGA cohort as well. One limitation of the in silico analysis is that few *NOTCH1* mutations have been functionally characterized as we did for C478F.

In T-ALL cases (Figure 3), we found many more missense/INDEL mutations in the HD domain than expected by chance (n = 768 out of 843) (35), which was consistent with previous studies demonstrating that missense and in-frame mutations in this domain activate NOTCH1 signaling [25,32,46]. Because a disproportionally large number of observed *NOTCH1* mutations in T-ALL cases mapped to the HD domain (92%), it skewed the estimated number of mutations expected to be randomly distributed over the remaining regions. To offset this potential confounding effect, we used the remaining 75 observed mutations to estimate the expected number in the other *NOTCH1* domains in T-ALL cases plotted in Figure 3. We found enrichment in *NOTCH1* missense mutations in the PEST and the Lin-12/NOTCH repeat 3 (LNR3) domains relative to the expected numbers, which was consistent with reports that mutations in these regulatory regions lead to NOTCH1 activation in hematological cancers [48]. A more precise algorithm that takes these and other confounding issues into account while predicting the likely functional impact of *NOTCH1* mutations in cancer patients is currently under development by our group (manuscript in preparation).

### 3.4. Evidence of Activating NOTCH1 Mutations in HNSCC

Activating *NOTCH1* mutations are largely absent from HNSCCs based on more than 500 TCGA patient tumors analyzed and from the majority of other genomic studies in which *NOTCH1* was sequenced, including patient cohorts in Japan, India, Europe, Latin America, or the United States (Table 1). However, two independent groups examining HNSCC in Chinese patients [16,17] reported *NOTCH1* mutations that were potentially activating alongside the usual profile of LOF mutations. Specifically, the localization of missense mutations to three distinct *NOTCH1* regions was considered evidence for receptor activation, including alterations to the distal EGF repeats (i.e., EGF 24-EGF 29, Abruptex region); the LNR/HD region; and the TAD/PEST domains. Unlike the more upstream EGF-like repeats (e.g., EGF 8–12) that are implicated in NOTCH1 ligand binding, the more distal EGF repeats form the Abruptex region that may prevent certain negative effects of ligands in *Drosophila* [49]. One interpretation is that missense mutations in the Abruptex region would mimic NOTCH1 signaling by relieving such negative regulation [16]. However, the functional impact is more complicated because strong mutations in the Abruptex region can also reduce NOTCH1 function [49]. In fact, one of the scientists who originally reported a hotspot mutation (C1133Y) in the *NOTCH1* Abruptex region of Chinese HNSCC patients [16] subsequently cloned the mutation and discovered that it prevented expression of NOTCH1 at the cell surface and a complete loss of canonical NOTCH1 signaling [50]. Therefore, missense mutations observed in the *NOTCH1* Abruptex region may actually be inactivating. A total of 18 potentially activating missense mutations in the *NOTCH1* HD domain have been identified in tumors from Chinese HNSCC patients. The majority of these were reported by Song et al., who identified two recurrent hotspot amino acid mutations (P1641S/L and N1713D) found in 10 patients [16]. Two other studies also observed distinct HD domain mutations in their cohorts including Chinese HNSCC patients [17,36]. HD and PEST domain missense mutations are prevalent in T-ALL, where they lead to increased NOTCH1 signaling, driving tumor formation. Therefore, we compared the HD and PEST domain mutations in T-ALL from the COSMIC database with those reported for Chinese HNSCC patients by examining their PROVEAN and SIFT scores. Mutations with PROVEAN scores ≥ −2.5 or SIFT values ≥ 0.05 are predicted to be neutral or tolerated (i.e., non-impactful), whereas mutations scoring below these thresholds are predicted to be more deleterious/damaging (i.e., impactful) with increasing confidence as the scores decrease. In T-ALL, 85% of HD mutations and 66% of PEST mutations observed were predicted to be impactful compared to just 44% and 28% of HD and PEST mutations found in Chinese HNSCC patients (Figure 4).

It is widely accepted that mutations occur in a more or less random fashion in tumors, but there is selective outgrowth (i.e., over-representation) only when a mutation conveys some advantage. If there is no advantage, then both non-impactful and impactful mutations will occur at theoretical frequencies that can be calculated based on the codon usage of a gene, known frequency of the different types of DNA base pair changes in a cancer type, and pre-calculated PROVEAN and SIFT scores for all possible mutations obtainable from single base pair substitutions. There are 172 codons in the HD domain and 180 codons for the PEST domain. Estimating the cancer type-specific frequencies of DNA base pair changes from whole exome sequencing in T-ALL [51] or the HNSCC TCGA, we calculated the theoretical frequencies expected for impactful and non-impactful mutations in the HD and PEST domains, assuming a random mutation model, and compared them to what was observed (Appendix A). In T-ALL, impactful mutations occurred at a rate 2.5- to 3-fold higher than expected by chance (*p* < 0.0001), consistent with mutations in these regions providing a driver function. In HNSCC, however, the number of observed mutations predicted to be impactful for both the HD and PEST domains were not significantly different from what was expected by chance (*p* = 0.12 and *p* = 0.67), increasing the likelihood that mutations observed from these domains in HNSCC were passengers rather than oncogenic drivers.

Next, we performed a more granular analysis of the differences between *NOTCH1* HD domain mutations in T-ALL and HNSCC patients (all studies). As shown in Figure 5A, we plotted each missense mutation according to its PROVEAN score (X-axis) and log10 transformation of its SIFT score (Y-axis). The PROVEAN scores for the mutations ranged from −13 to 4, and the log10 SIFT scores ranged from −6 to 1, with mutations predicted to be the most disruptive mapping to the lower left quadrant and alterations least likely to change protein function mapping to the upper right quadrant. Hotspot mutations observed in multiple patients are represented by larger circles, with the diameter of each circle proportional to the number of patients with a specific cancer type having the mutation.

Many hotspot HD domain *NOTCH1* mutations in the T-ALL patients mapped to the lower left quadrant, indicating a high probability that their amino acid substitutions altered protein function. In sharp contrast, most of the HD domain mutations in the Chinese HNSCC patients, including reported hotspot substitutions, were either in the upper right quadrant (i.e., likely neutral) or greatly shifted in that direction. Lastly, we examined the distribution of missense mutations with respect to amino acid position within the HD domain (Figure 5B). The majority of the T-ALL mutations predicted to be deleterious were in three distinct clusters, whereas the mutations in the Chinese HNSCC patients were outside these clusters and topologically overlapped the T-ALL mutations predicted to be neutral.

Collectively, these analyses suggest striking differences between the *NOTCH1* HD domain missense mutations in T-ALL and Chinese HNSCC patients, decreasing the likelihood that the mutations in the latter group are strongly activating. We cannot rule out the possibility that they may represent much weaker activating mutations. Some support for this comes from the missense mutations occurring in the LNR3 domain (immediately proximal to the HD domain region) found in four Chinese patient tumors from the study by Izumchenko et al. [42], which are predicted to be deleterious by their PROVEAN/SIFT values (not shown). Mutations in the HD domain far outnumber those in the LNR3 domain by two orders of magnitude in T-ALL, despite the only fivefold difference in size, further supporting that the latter may produce a weaker phenotype. While evidence for inactivating *NOTCH1* mutations in HNSCC is abundant, the argument for complex activating *NOTCH1* mutations in this disease is markedly less robust.

## 4. NOTCH1 Signaling in HNSCCs

### 4.1. HNSCC Cell Line Models for Studying NOTCH1

Our group and others have comprehensively profiled the genetic alterations in established HNSCC cell lines [52,53,54] and found that somatic mutations and DNA copy number changes faithfully recapitulate the genomic landscape and driver events identified for primary HNSCC tumors. WES of 56 HPV-negative and nine HPV-positive established HNSCC cell lines identified a total of 19 nonsynonymous *NOTCH1* mutations among 17 cell lines (Appendix A) [12,15,52], including two that were also HPV positive. Roughly, 25% of HPV-negative cell lines examined contained a *NOTCH1* mutation and most were homozygous alterations predicted to be inactivating. We confirmed total loss of NOTCH1 protein expression by Western blotting in a subset of cell lines with homozygous truncating mutations [12]. An absence of canonical NOTCH1 signaling (i.e., ICN1) before and after stimulation with the NOTCH1 ligand JAG1 was also confirmed in HN31 [12], which harbors a homozygous point mutation in the EGF 12 domain. The expression of total NOTCH1 and activated intracellular NOTCH1 (ICN1) protein was examined in a large panel of our HNSCC cell line panel with reverse phase protein arrays (RPPAs). Antibodies that recognize the C-terminal domain of NOTCH1, regardless of cellular location, or only ICN1 (i.e., Cl-NOTCH1) were validated with the RPPA platform using control *NOTCH1*-wt and mutant *NOTCH1* cell lines. A range of expression for both total NOTCH1 and ICN1 was identified by RPPA that correlated well with immunoblot results when examined in a subset of cells (Table 2). Levels of ICN1 measured by RPPA were significantly higher for cell lines harboring wt *NOTCH1* with immunoblot evidence of elevated ICN1 (Figure 6A) compared to *NOTCH1*-wt cells with lower ICN1 expression by immunoblotting (*p* < 0.05) or to *NOTCH1* mutant cell lines (*p* < 0.001). Collectively, the RPPA and Western blot data indicate that roughly two thirds of *NOTCH1*-wt HNSCC tumor lines have little to no baseline activation of the NOTCH1 pathway (Figure 6B).

### 4.2. Restoration of *NOTCH1* Signaling Alters Growth of NOTCH1 Mutant HNSCC Cell Lines

We developed multiple cell line models for examining the functional consequences of NOTCH1 signaling in HNSCC cases. Initially, the activated form of wt NOTCH1 (i.e., ICN1) was stably re-expressed in the *NOTCH1* mutant HNSCC cell lines HN4, PCI-15B, UMSCC-47, and HN31 using a green fluorescent protein-tagged bicistronic retroviral vector (MigR1-ICN1). In vitro viability and proliferation of *NOTCH1* mutant cells rapidly declined after over expressing ICN1 [12]. To model a more physiological response, a full-length wt *NOTCH1* gene was subcloned into MigR1 and expressed in *NOTCH1* mutant cell lines where it was found to significantly impair proliferation in cells grown on JAG1 or injected into mice [12]. These experiments demonstrated the tumor-suppressive capacity of NOTCH1 signaling in naturally occurring *NOTCH1* mutant cell lines.

### 4.3. *NOTCH1* Pathway in HNSCCs Harboring wt NOTCH1.

A recent study by Loganathan et al. suggests that NOTCH pathway inactivation is common in HNSCC even when the canonical pathway components are not directly affected [57]. They studied the significance of recurrent but infrequently mutated genes (“long tail” mutations) in HNSCC using an in vivo CRISPR screen of 484 genes to identify those that lead to cancer formation when mutated in various, relevant oncogenic backgrounds. The most prevalent tumor suppressors identified were ADAM10, AJUBA, receptor interacting serine/threonine kinase 4 (RIPK4), NOTCH2, and NOTCH3, which were all shown to converge on the NOTCH pathway. They estimate that 67% of human HNSCC patients have NOTCH pathway inactivation—including about 27% with inactivating *NOTCH1/2/3* mutations and about 40% with inactivating alterations of *ADAM10*, *RIPK4*, or *AJUBA* or amplification of *NUMB*, an AJUBA interactor. The evidence in this study strongly supports the tumor suppressive role of the NOTCH pathway in HNSCC.

Our in vitro data support Loganathan et al.’s conclusion. As described above, a number of HNSCC cell lines express NOTCH1-wt protein but have low basal pathway activation defined by relatively low expression of ICN1 when probed with ICN1 (anti-Cl-NOTCH1) antibodies. For example, basal levels of activated NOTCH1 are much lower in PJ34 compared to FaDu cells (Figure 7A), which may explain why the latter has frequently been used to study NOTCH1 function in HNSCC [58,59,60]. Cultivation of *NOTCH1*-wt HNSCC tumors harboring low basal levels of ICN1 on immobilized JAG1 can trigger NOTCH1 activation, as evident by the strong induction of ICN1 detectable by ICN1 antibody (Figure 7B). We examined the phenotype of PJ34 after prolonged growth on JAG1. Subtle changes in cell morphology accompanied by slower growth were observed on day 3, and by day 5, individual cells had shrunk profoundly in size and formed loosely attached spheres (Figure 7C).

Previously, we reported that stimulation of the NOTCH pathway in *NOTCH1*-wt HNSCC cell lines led to significant downregulation of two prominent proto-oncogenes, AXL and CTNNAL1 (α-catulin) [56], which have both been linked to aggressive biologic and clinical features in HNSCC and other tumor types [61,62,63]. This is further supported by our RPPA data from cell lines that demonstrated a strong inverse correlation between levels of ICN1 and total AXL protein (Figure 6C). To further understand the phenotypic change in PJ34 cells grown on JAG1, we examined alterations in gene expression on day 5, when the morphologic changes peaked. In addition to AXL and CTNNAL1, hundreds of genes were significantly upregulated or downregulated after growth on immobilized JAG1, compared to growth on immobilized control (FC) protein. We focused our analysis on genes regulating adhesion and markers of stem cells, epithelial-to-mesenchymal transition (EMT), and markers of early differentiation (Appendix A). Downregulation of laminin chain gamma 2 (LAMC2) and laminin chain alpha 3 (LAMA3) involved in integrin adhesion, along with integrin alpha 5 (ITGA5), ITGA3, ITGA6, and integrin beta 6 (ITGB6), was observed. In contrast, the early differentiation markers keratin 4 (KRT4) and KRT13 were upregulated by NOTCH activation. Only one EMT marker was upregulated by JAG1 exposure, while two common markers of EMT, vimentin (VIM) and fibronectin 1 (FN1), went down. Two putative stem cell markers, SRY-box transcription factor 2 (SOX2) and aldehyde dehydrogenase 3 A1 (ALDH3A1), were upregulated after NOTCH activation, but most of the other common stem cell markers were unchanged. Perhaps because of the late time point, most canonical downstream HES/HEY family members were unchanged, but HES5 remained significantly elevated.

### 4.4. Evidence That NOTCH1 Is Oncogenic in a Subset of HNSCCs

In contrast to our findings, a number of publications have presented evidence that NOTCH1 signaling is oncogenic, or at least may have a dual role, in a subset of HNSCC [16,17,64]. Four lines of experimental evidence to support a putative oncogenic function for NOTCH1 have been pervasive in this body of literature: (1) manipulation of NOTCH1 signaling in HNSCC cell lines alters tumor spheroid growth in vitro; (2) knockdown of total NOTCH1 expression with short hairpin RNA (shRNA) leads to reduced growth in vitro; (3) NOTCH1 expression and activation correlates with expression of cancer stem cell markers in vitro; (4) NOTCH1 activation can occur in primary HNSCC tumor samples and in some cases correlates with aggressive disease. Evidence for each of these observations is described below, along with an explanation of possible limitations.

Multiple authors have reported that blocking *NOTCH1* function with shRNA leads to a decrease in HNSCC tumor spheroid growth in three-dimensional (3-D) cultures, and conversely expression of ICN1 can increase survival and growth of spheroids or enhance tumor growth in mice [65,66,67]. Although the ability to survive as unattached cells is required for 3-D growth, the property in itself is not sufficient to define cancer stem cells. In fact, the bulk of an actively growing tumor spheroid likely comprises transiently amplifying progenitors. The gold standard for quantifying cancer stem cells (i.e., tumor initiating cells) is the limiting dilution inoculation assay in mice. No published studies we are aware of have functionally examined the impact of NOTCH1 signaling on cancer stem cell frequency in mouse tumor models in the absence of pre-cultivation as spheroids. This makes it difficult to distinguish tumor-initiating properties from in vitro survival in the detached state, or anoikis resistance.

Knocking down total NOTCH1 expression with small interfering RNA (siRNA) or shRNA has also been reported to inhibit proliferation in regular cultures for a handful of HNSCC cell lines with wt *NOTCH1* expression [17,65]. The level of inhibition in published studies is variable, with only modest growth effects in some cases. Furthermore, parallel experiments employing drugs that target NOTCH1 activation (i.e., γ-secretase inhibitors) in some of the same cell lines did not always reveal dramatic growth inhibition at concentrations sufficient to block production of ICN1 [17,65]. There are multiple caveats to interpreting these kinds of experiments. Isolation of stable long-term shRNA clones can sometimes produce clonal variation in proliferation rates that is artifactual and independent of the gene being knocked down. Because NOTCH1 receptors compete with NOTCH2 receptors for the same ligands in the context of cis-inhibition (i.e., co-expression on the same cell), it is unknown how changing this dynamic could impact NOTCH2 signaling in cells where *NOTCH1* is knocked down. In our hands, concentrations of γ-secretase inhibitors that block NOTCH1 signaling have no impact on in vitro growth or survival for HNSCC cell lines, even in tumors that normally express high basal levels of ICN1.

Several investigators have found that NOTCH1 expression and activation correlate with the expression of cancer stem cell markers [65,66,68], including SOX2 and aldehyde dehydrogenase (ALDH) activity measured by the ALDEFLUOR assay. Our laboratory confirmed that SOX2 and ALDH3A1 mRNA was strongly upregulated after NOTCH1 signaling in vitro (Appendix A). However, ALDH3A1 is not solely a marker of basal stem cells as it is also expressed significantly in the suprabasal layers of normal mucosa [69]. Moreover, Kato et al. demonstrated that normal human oral keratinocytes that were strongly positive for aldehyde dehydrogenase activity, using the ALDEFLUOR assay typically used to enrich for HNSCC cancer stem cells, were actually enriched for keratinocytes that had undergone early differentiation [69]. Although SOX2 is frequently mentioned as a stem cell marker, immunohistochemistry (IHC) studies have demonstrated SOX2 expression in both the basal layer and immediate suprabasal layer of normal squamous mucosa [70], implying a role in early differentiation too. Evidence against SOX2 as a cancer stem cell marker comes from the report that elevated SOX2 expression in HNSCC is actually associated with a better clinical outcome in this disease [70], while loss of SOX2 expression conversely correlates with poor survival [71]. Thus, it is unclear if the associations between NOTCH1 and expression of ALDH3A1 and SOX2 represent a genuine shift toward a cancer stem cell phenotype, or instead signify the transition to early differentiated progenitors (i.e., transiently amplifying cells).

Activation of NOTCH1 is thought to trigger an asymmetric division of the basal stem cell in squamous epithelium to produce more differentiated migratory daughter cells [72,73]. Conditional knockout of *NOTCH1* in mouse squamous epithelium leads to an expansion of cells expressing basal markers KRT14, ITGB1, and ITGB4 [74]. When NOTCH1 activation occurs in basal stem cells, they likely migrate outwards and away from the basement membrane due to loss of integrins induced by NOTCH1 signaling [74]. Studies in mouse esophageal epithelium have demonstrated that loss of NOTCH1 signaling, rather than activation, leads to localized clonal expansion of basal stem cells harboring defective *NOTCH1* that outcompete neighboring stem cells with NOTCH1 signaling still intact [75]. Additional evidence demonstrating the selective advantage and clonal expansion of basal stem cells harboring *NOTCH1* LOF mutations comes from mathematical modeling in a study that used deep sequencing data from non-cancerous sun-exposed skin in humans [76]. Spatial analysis of mutations from multiple regions of skin in that study indicated a selective advantage for stem cells harboring disruptive *NOTCH1* mutations. Mechanistically, this explains how loss of *NOTCH1* drives early carcinogenesis of HNSCC, by allowing expansion and persistence of stem cells that eventually accumulate additional genomic alterations.

We have observed that integrins and extracellular matrix (ECM) proteins normally associated with maintaining basal stem cell attachment are downregulated in HNSCC after NOTCH1 activation in vitro. The basal stem cell marker keratin 14 (KRT14) was downregulated by NOTCH1 signaling, while the early differentiation markers KRT13 and KRT4 are conversely upregulated by NOTCH1 activation in vitro (Appendix A). Collectively, our data support the idea that activation of NOTCH1 in HNSCC recapitulates its normal biological function to turn on a program of gene expression associated with very early differentiation. This would produce a phenotype resembling a transiently amplifying progenitor, rather than an actual cancer stem cell. Expression of ICN1, SOX2, and ALDH3A1 in the suprabasal layer of normal tissue [64,69,70] is consistent with such a model.

### 4.5. *NOTCH1* Pathway Activation in Clinical HNSCC Specimens

Both direct and indirect evidence demonstrate that NOTCH1 signaling can occur in a subset of tumors from patients with HNSCC. Multiple groups have inferred the NOTCH1 pathway status in HNSCCs from expression of NOTCH1 pathway genes/proteins and their putative downstream targets. Expression of total NOTCH1 and/or its ligands (JAG1, JAG2, DLL1, and DLL4) is frequently higher in HNSCC samples than in adjacent nonmalignant mucosa samples [17,65,77]. However, ligand and receptor expression may not lead to increased NOTCH1 activation. Ordinarily, NOTCH1 signaling takes place in a juxtacrine manner with ligand expressed on the surface of neighboring cells binding to and activating NOTCH1 on a proximal cell. In contrast, interaction of NOTCH1 with ligand expressed on the same cell can prevent NOTCH1 signaling through a process known as cis-inhibition [78].

Researchers have also used expression of HES and HEY family members, which are known downstream targets of NOTCH1, as a surrogate for NOTCH1 activation in human HNSCCs [17,65]. However, just as the same NOTCH ligands can signal and bind multiple NOTCH family receptors, some of the downstream HES/HEY targets are also shared. We previously reported that short-term activation of NOTCH1 leads to measurable increases in *HES2*, *4*, *5*, and *HEY1* and *2* mRNA [55]. Increases in HES/HEY are effects which are likely transient and cyclic. When PJ34 was grown on JAG1 ligand for an extended period (i.e., 5 days), only *HES5* was significantly elevated (Appendix A)—although there was a trend towards persistent increased expression of *HEY1* that did not reach significance. We compared HNSCC cell line models (Table 2) with demonstrable evidence of high basal activation of NOTCH1 by immunoblotting (n = 5) (Table 2) to cell lines with documented homozygous inactivating *NOTCH1* mutations (n = 6) to examine if RNA expression of any *HES/HEY* family members were reliable biomarkers of chronic NOTCH1 signaling. None of the HES/HEY family members showed significantly different expression between the two groups of cell lines (Figure 8), although like PJ34 grown on JAG1, there was a non-significant trend towards elevated *HES5 and HEY1*. Consequently, efforts to use expression of *HES/HEY* genes as a surrogate for NOTCH1 pathway activation in HNSCC may lack robustness.

Rettig et al. [64] used an antibody that only recognizes ICN1 to directly examine NOTCH1 activation in a cohort of archival HNSCC tumors by IHC. They found two very distinct patterns of ICN1 staining in a subset of samples positive for ICN1. In the peripheral pattern, ICN1 was apparent in just a single layer of tumor cells positioned immediately behind the outermost advancing edge of tumor cells that stained negative at the stromal interface. In the non-peripheral or more diffuse pattern, ICN1 expression was visible in a fraction of tumor cells scattered throughout tumor nests. Notably, non-peripheral expression of ICN1 occurred in 34% of samples, from multiple anatomical subsites. Given the small percentage of tumor cells with NOTCH1 activation in samples with the peripheral pattern, it is unlikely that any RNA expression analysis of whole tumor lysates could identify this subset. In a follow-up study, Rettig et al. [79] used IHC to examine correlations between staining for ICN1, HEY1, and JAG1 in a cohort of HNSCC. Three expression patterns were noted for all three molecules, which again included peripheral only tumor staining, diffuse staining scattered throughout tumor nests, or no staining. While there was a strong correlation between non-peripheral expression of JAG1 and ICN1, it was not true for HEY1. Consequently, intra-tumoral JAG1 expression probably can stimulate NOTCH1 activation in a subset of HNSCC tumors, but HEY1 expression on its own may not be a reliable marker for NOTCH1 pathway activation in HNSCC.

The fact that NOTCH1 signaling takes place in a subset of HNSCC cells is not proof of an oncogenic role. It is also possible that HNSCCs with persistent NOTCH1 signaling have adapted to minimize the potential anti-growth effects of the pathway through acquisition of compensatory genomic alterations. Ultimately, NOTCH1 triggers a program of genes involved in early differentiation that may have different phenotypic consequences depending on a tumor’s genetic background. Presently, clear evidence that HNSCCs with intra-tumoral NOTCH1 activation have a selective advantage is lacking.

### 4.6. Epithelial to Mesenchymal Transition and *NOTCH1* Signaling

Epithelial-to-mesenchymal transition (EMT) is crucial for cell differentiation and morphogenesis during embryonic development. Cancer cells that undergo EMT can invade and metastasize to distant sites. The NOTCH pathway has been implicated in EMT in many human cancers [80], including HNSCC, where several different pathways downstream of NOTCH have been proposed. Activated NOTCH signaling may promote tumor metastasis by 3-phosphoinositide-dependent protein kinase 1 (PDK1)-induced EMT in hypopharyngeal cancer cells [81], or Snail-mediated EMT progression in oral cavity squamous carcinoma cells [82]. Likewise, in tongue cancers, NOTCH1 activation results in a mesenchymal phenotype, through increased focal adhesion kinase (FAK) activity in a PTEN-RBP-Jκ-dependent manner, which was inhibited upon NUMB overexpression [83]. Moreover, NOTCH1 depletion in a HNSCC cell line led to c-Myc downregulation and thereby reduced EMT and invasion [84]. Other NOTCH signaling components were also shown to impact the EMT phenotype, with the NOTCH4-HEY1 pathway promoting EMT, leading to increased invasion and migration in HNSCC. Additionally, signaling cross-talk in squamous cell carcinoma involving NOTCH1 with the transforming growth factor beta (TGF-β) pathway represses NOTCH3 through zinc finger E-box binding homeobox 1 (ZEB1) and promotes EMT and tumor initiation [85]. Collectively, these data imply a direct connection between NOTCH pathway activation and EMT in HNSCC. In contrast with the body of literature cited above, our group has not found a robust connection between the NOTCH1 pathway and EMT in HNSCC cell lines where NOTCH1 signaling is manipulated in vitro. Chronic in vitro NOTCH1 activation in the HNSCC cell line PJ34 actually decreased expression of EMT markers (Appendix A), which was also observed in a second *NOTCH1*-wt tumor line (manuscript in preparation).

## 5. *NOTCH1* Mutations in Other SCCs

SCCs are the most common ectodermal cancers which originate from squamous cells in the head and neck mucosa, lungs, skin, esophagus, and cervix. NOTCH pathway alterations have been implicated in many of these SCCs, with *NOTCH1* being the most commonly mutated gene besides *TP53* [20]. Cutaneous SCC (cSCC) is the second most prevalent skin cancer worldwide. WES analysis revealed LOF mutations in *NOTCH1* in ~75% of cSCC (primary cSCC and cSCC cell lines) [22]. Furthermore, genomic analysis of aggressive cSCC showed that 59% of cases have *NOTCH1* mutations, of which more than 30% were LOF [86]. These mutations, as in HNSCC, appeared to be clustered at the N-terminal domains of *NOTCH1*. Moreover, targeted sequencing of a metastatic cSCC cohort identified frequent LOF mutations (24%) and copy number variations (CNVs, 69%) of unknown functional significance in *NOTCH1* [87].

The genomic landscape of esophageal SCC (ESCC) is comparable to that of HNSCC. An earlier study reported *NOTCH1* LOF mutations in 21% of North American ESCCs and 2% of Chinese ESCCs [88]. However, a subsequent study identified more frequent mutations in *NOTCH1* in Chinese ESCC (13%) using WES [89]. Another independent study published in the same year reported *NOTCH1* mutations (INDELs, SNVs, and amplifications) in 9.1% of Chinese ESCC cases [90]. A study of Japanese ESCC patients [91] found LOF *NOTCH1* mutations in 18.6% of ESCC. Notably, *NOTCH1* mutations in ESCC were confined to the N-terminal domains and predicted to cause LOF as in other SCCs.

Likewise, *NOTCH1* was mutated in 8% of lung SCC from the TCGA dataset, most of which were truncating mutations similar to those found in HNSCC [14]. *NOTCH1* mutations are found in roughly 7% of cervical squamous cell carcinoma (CSCC) TCGA patient samples [33]. Excluding missense mutations predicted to be tolerated by either SIFT or PROVEAN, roughly one third of the CSCC mutations found in the COSMIC database are truncating and predicted to be LOF, with some missense mutations localizing to EGF domains also predicted to cause inactivation. Two of the CSCC *NOTCH1* missense mutations that localize to the HD domain resemble those reported for HNSCC Chinese patients in that their combined SIFT and PROVEAN scores do not suggest a potent phenotype. Taken together, *NOTCH1* likely has a tumor suppressor function in a wide range of SCCs.

## 6. The Prognostic Role of NOTCH Signaling and *NOTCH1* Mutations in HNSCC

The prognostic role of the NOTCH pathway in different subsets of HNSCC has been under scrutiny in recent years. Whether NOTCH1 signaling promotes or suppresses tumor formation, progression, and metastasis for various HNSCC types is an open question. Some of the studies performed to address this issue are presented below. However, several caveats exist which may explain the disparity of results often reported in the literature. As we have addressed in this review, not all *NOTCH1* mutations observed in cohorts are predicted to be impactful and few studies have attempted to make this distinction. The heterozygous verses homozygous status of *NOTCH1* LOF is frequently difficult to know in analyses of clinical specimens where contaminating normal tissue can be significant. For example, Rettig et al. found ICN1 staining by IHC in HNSCC tumor samples from patients with truncating *NOTCH1* mutations [64], presumably due to a remaining wt allele. Many reports are also likely underpowered due to the small numbers of *NOTCH1* mutations in the studies and the unavailability of suitable validation cohorts makes the conclusions difficult to generalize. When only the *NOTCH1* mutational status is considered in studies, the proportion of *NOTCH1*-wt patients in the comparison arm that may have the NOTCH1 signaling pathway active, or silenced through other means, is likely to be variable and could confound the clinical behavior of tumors otherwise grouped together for analysis.

Mutations in *NOTCH1* were strongly associated with shorter overall (OS) and disease-free survival (DFS) (*p* = 0.004 and *p* = 0.001*,* respectively) in Chinese oral squamous cell carcinoma (OSCC) patients compared to patients with wt *NOTCH1* [16]. A second study had similar results, in that *NOTCH1* mutant OSCC (oral tongue) had significantly poorer outcomes, including shorter DFS (*p* = 0.005) and patients were more likely to die with recurrent disease compared to those without mutations [92]. Contradictory to these reports, Japanese OSCC patients with *NOTCH1* mutations showed a longer median DFS (*p* < 0.05) in comparison to the patients with wt *NOTCH1* [37]. A similar trend was observed in patients with stage IV HNSCC (oropharynx and hypopharynx), with mutated *NOTCH1* significantly correlating with improved OS (*p* = 0.04) [93].

In addition to *NOTCH1* mutations, studies of NOTCH1 protein expression levels also showed mixed clinical outcomes in HNSCC [58,94]. While overexpressed NOTCH1 protein levels correlated with improved OS (*p* < 0.001) in patients with oropharyngeal cancer [94], it was an independent prognostic factor for poor OS (*p* = 0.015) and correlated with distant metastases (*p* = 0.003) and tumor differentiation (*p* = 0.031) in patients with hypopharyngeal squamous cell carcinoma [58]. Both of these studies used an anti-NOTCH1 antibody that recognizes intracellular epitopes and therefore total NOTCH1 as well as ICN1. Additionally, elevated NOTCH1 protein expression was associated with resistance to chemotherapy in HNSCC. Patients lacking NOTCH1 protein exhibited improved response to neoadjuvant chemotherapy, with better overall (*p* < 0.05) and progression-free survival (*p* < 0.05) compared to patients expressing NOTCH1 protein [95]. Rettig et al. [64] examined clinical characteristics with the presence and pattern of ICN1 staining in HNSCC tumors that were wt for *NOTCH1* in archival specimens. They found that tumors with a non-peripheral pattern of ICN1 expression were more highly associated with extracapsular spread compared to tumors with a more peripheral pattern of NOTCH1 activation, although the cohort of wt *NOTCH1* tumors analyzed for clinical correlations was predominately HPV positive.

The potential impact of NOTCH signaling and its components has also been studied in HNSCC subtypes. NOTCH pathway upregulation as analyzed from gene expression analysis, using the NanoString platform and the PanCancer Pathways Advanced Analysis Module, significantly correlated with cancer-specific mortality (*p* = 0.032) in OSCC patients [96]. The downstream targets of the NOTCH pathway, HES1 and HEY1 expression levels, and their association with clinical outcomes were analyzed in sinonasal squamous cell carcinoma [97] and HNSCC [79], respectively. Although higher HES1 mRNA levels were associated with better OS (*p* = 0.04), elevated HEY1 protein expression was associated with worse OS (*p* = 0.009) and DFS (*p* = 0.001). However, extrapolation of the NOTCH pathway through its downstream targets needs to be made with caution given the potential cyclic nature of expression and the caveats discussed in detail in Section 4.5.

Moreover, clinical impacts of NOTCH1 mRNA levels have also been ascertained in HNSCCs. Higher NOTCH1 gene expression was associated with better OS (*p* = 0.013) and DFS (*p* = 0.040) in HNSCC tissues [98], and longer DFS (*p* = 0.039) in laryngeal cancer [99].

## 7. Targeting *NOTCH1* Mutant HNSCC

Developing targeted therapies for HNSCC has remained a challenge given the heterogeneity of the tumor landscape, which is mostly dominated by tumor suppressors rather than oncogenes. Various treatment strategies for HNSCC have had limited success, owing to low response rates, acquired resistance, and frequent relapse. Currently biomarker-driven therapies for *NOTCH1* mutant HNSCC are lacking. Because activating *NOTCH1* mutations are largely absent from HNSCCs, therapies that target NOTCH are not useful for HNSCC. These therapies include γ-secretase inhibitors, an inhibitor of the NOTCH transcription complex (CB-103), and antibodies to DLL-4, NOTCH1, and NOTCH2/3 [100].

### 7.1. PI3K Inhibitors

The PI3K/Akt/mammalian target of rapamycin (mTOR) signaling cascade governs critical cellular processes, including cell growth, proliferation, metabolism, and motility, all of which are required by diverse human cancers to survive, grow, and metastasize. As a result, this pathway is activated in cancer cells through various mechanisms involving genomic alterations in the pathway components (e.g., PIK3CA, PIK3R1, PTEN, AKT, TSC1, TSC2, LKB1), some of which are therapeutic targets. Pathway inhibitors include drugs that target PI3K, AKT, mTOR, or a combination of them [101]. Several PI3K inhibitors are currently under clinical development for treatment of different cancers; of these, copanlisib and alpelisib have been approved by the U.S. Food and Drug Administration (FDA) for treatment of hematological malignancies [101,102,103]. However, treatment of solid tumors with PI3K inhibitors has resulted in only modest responses.

More than 90% of HNSCCs have an upregulated PI3K/AKT/mTOR pathway which can be attributed to EGF receptor activation, PI3K overexpression, *PIK3CA* mutations and amplifications, and *PTEN* LOF mutations. HNSCCs had varied responses to treatment with PI3K inhibitors, ranging from inhibition of tumor cell proliferation to radio-sensitization in both in vitro and in vivo models. However, the existence of feedback pathways ultimately leads to resistance to PI3K pathway inhibition, posing a challenge in the sustained treatment of this cancer [104].

Previous research in our laboratory showed that PI3K pathway inhibition in HNSCCs induced growth arrest in *PIK3CA* mutant tumors [105], and significant cell death in *NOTCH1* mutated tumors [15]. In vitro studies of HNSCC cell lines harboring *NOTCH1* LOF mutations demonstrated sensitivity to PI3K/mTOR inhibitors, resulting in apoptosis and reduced clonogenic growth compared to cell lines with wt *NOTCH1*. Similarly, PI3K/mTOR inhibition showed markedly increased apoptosis and impaired tumor growth in *NOTCH1* mutant HNSCC xenograft models. Moreover, genomic depletion of *NOTCH1* sensitized most wt *NOTCH1* HNSCC cell lines to PI3K/mTOR inhibitor-mediated apoptosis.

Further mechanistic investigation of these findings revealed several differentially expressed proteins in *NOTCH1* mutant HNSCC cells upon PI3K/mTOR inhibition compared to *NOTCH1*-wt cells. One of these proteins is PDK1, a downstream signaling molecule of the PI3K pathway, which was significantly downregulated upon PI3K/mTOR inhibition exclusively in *NOTCH1* mutated HNSCC cell lines. In addition, depletion or inhibition of total PDK1 in *NOTCH1*-wt HNSCC sensitized them to PI3K/AKT inhibition, resulting in apoptosis.

These preclinical findings are encouraging and are supported by the results of three other independent studies. The potent PI3K inhibitor PX-886 significantly reduced tumor growth in two *NOTCH1* mutant HNSCC patient-derived xenograft models [106]. Likewise, activated Notch signaling conferred resistance to PI3K/mTOR inhibitors to breast cancer [107]. A phase I study with the dual PI3K/mTOR inhibitor bimiralisib (PQR309) was conducted in patients with heavily pretreated advanced solid cancers. One HNSCC patient with a *NOTCH1* LOF mutation had a partial response (85% reduction in the target lesion) on bimiralisib treatment that was sustained for 36 weeks [108]. Currently, this observation is being validated in a phase II study to test the efficacy of bimiralisib treatment in *NOTCH1* mutant HNSCC patients (NCT03740100).

Collectively, these findings suggest that *NOTCH1* inactivation predicts the response of HNSCC to PI3K inhibition and may lead to the development of biomarker-driven therapy for HNSCC.

### 7.2. Chemotherapy

The current standard of care therapy for HNSCC includes cisplatin given concurrently with radiotherapy as a primary treatment or after surgery. However, all patients experience adverse effects and many develop resistance to chemoradiation, resulting in recurrence and metastasis. Therefore, understanding the mechanisms underlying the development of chemoresistance of HNSCC is critical. A comprehensive protein expression analysis predicting chemoresistance of HNSCC cells revealed that increased total NOTCH1 expression was associated with sensitivity to cisplatin-based treatment [109]. However, several other studies found a strong correlation between increased NOTCH1 expression and cisplatin resistance; therefore, the researchers advocated the use of a γ-secretase inhibitors to sensitize HNSCC to chemotherapy [95,110]. Furthermore, elevated levels of Stat3 and NOTCH1 expression were strongly associated with cisplatin resistance in HNSCC patients, which could be reduced by inhibiting Stat3 and NOTCH signaling [111]. Moreover, elevated NOTCH1 levels were significantly associated with chemotherapy-enriched cancer stem cell (CSC) populations, resulting in chemo-resistant HNSCC. NOTCH1 inhibition alone, or in combination with chemotherapy in this case, resulted in significantly reduced CSCs both in vitro and in vivo. Possibly, NOTCH1 signaling activates genes mediating early differentiation or transition to a transiently amplifying phenotype that include pro-survival or anti-apoptotic factors. This would be consistent with our observations that NOTCH1 activation increases anoikis resistance. Conceivably, in normal biology when basal stem cells undergo early differentiation through NOTCH1 activation, they detach from the basement membrane and would require additional or increased survival mechanisms. A similar process is likely to occur in HNSCC during the cascade of gene expression changes triggered by NOTCH1 signaling. The complex role of NOTCH1 activation in the tumorigenesis and phenotypic behavior of HNSCC would likely confound therapeutic strategies proposed to inhibit NOTCH1 signaling in this cancer type. Rather, the focus should be on elucidating and blocking pathways or survival mechanisms that might arise in subsets of HNSCC where NOTCH1 is activated.

### 7.3. Immunotherapy

Cancers evade immune surveillance through many mechanisms, such as a dysfunctional immune system, deficient expression of class I major histocompatibility complex molecules, secretion of cytokines, and expression and interaction of immune checkpoint molecules. The interaction of programmed cell death protein 1 (PD-1) and programmed death ligand 1 (PD-L1) on cell surfaces protects tumor cells from immune responses. Therefore, anti-PD-1/PD-L1 antibodies or PD-1/PD-L1 inhibitors have been tested for their therapeutic response in many cancers, including HNSCC, where two (pembrolizumab, nivolumab) are now FDA approved for recurrent or metastatic disease. A three-year observational study of 126 HNSCC patients predicted anti-PD-1/PDL-1 responses. Somatic frame shift mutations of common tumor suppressor genes (*NOTCH1* and *SMARC4*) were frequently observed in HPV-negative anti-PD-1/PD-L1 responders [112]. Furthermore, a more recent study identified *NOTCH1-3* LOF mutations as novel biomarkers predicting improved response to immune checkpoint inhibitor treatment in lung cancer patients [113]. Taken together, these studies suggest that *NOTCH1* mutation may be an important biomarker to predict response to immune checkpoint inhibitors. However, the underlying mechanism for this correlation is currently unknown.

## 8. Conclusions

About half of HNSCC patients die of the disease, making the development of more effective therapy for HNSCC an important goal. A challenge to developing targeted therapy for HNSCC is the dominance of mutations in tumor suppressors including *NOTCH1,* which is mutated in about 17% of HPV-negative HNSCC. Structural analysis of *NOTCH1* mutations in HNSCCs demonstrate that most are LOF mutations. This conclusion is supported by the distribution of truncations in HNSCC, where the majority occur before the TAD and likely lack regions required for ICN1 activity. *NOTCH1* missense mutations in HNSCCs are enriched in the EGF2, EGF4, EGF8, EGF9, EGF10, EGF11, EGF12, EGF29, RAM, and ANK1 domains in contrast to T-ALL, where *NOTCH1* mutations are activating and there are more mutations in the HD domain. These two distinct mutation distributions are non-overlapping, which support the model that *NOTCH1* mutations inactivate NOTCH1 signaling in HNSCC. However, one cannot rule out rare activating *NOTCH1* mutations. Additionally, manipulation of NOTCH signaling in *NOTCH1* mutant HNSCC cell lines demonstrates that it functions as a tumor suppressor in vitro and in vivo.

Studying NOTCH1 pathway activation is challenging because downstream events, such as the expression of HES and HEY family members, are likely transient and cyclic. Our own observations support the hypothesis that activation of NOTCH1 in HNSCC recapitulates its usual biological function to regulate a program of gene expression associated with very early differentiation, rather than EMT or CSC maintenance.

Currently, the presence of a *NOTCH1* mutation in HNSCCs does not affect clinical decision making, but this may change. Defining the prognostic role of the NOTCH pathway is challenging because of the aforementioned limitations in measuring NOTCH pathway activation, as well as the difficulty in determining if some *NOTCH1* mutations are passenger events or LOF mutations. As such, the clinical prognostic role of the NOTCH1 pathway status in HNSCC remains inconclusive. In regard to therapy, *NOTCH1* mutant, HPV-negative HNSCC may be more responsive to immune checkpoint therapy. In vitro, in vivo, and patient data support an ongoing clinical trial that is testing the hypothesis that HNSCCs with *NOTCH1* mutations are highly sensitive to PI3K inhibitors due to the ensuing loss of total PDK1 protein. Because *NOTCH1* LOF mutations are common in other SCCs, including those of skin, esophagus, cervix, and lung, these findings may have implications for the treatment of cancers beyond HNSCC.

## Figures and Tables

**Figure 1 cells-09-02677-f001:**
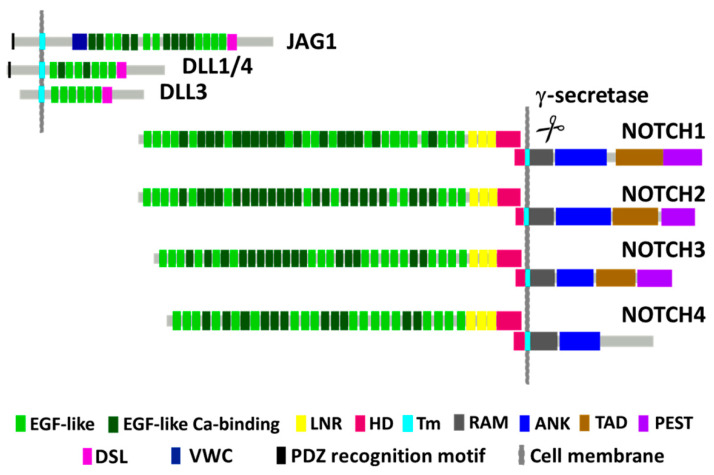
Schematic diagram of human NOTCH family member proteins and their ligands. NOTCH molecules are made as a larger precursor that gets cleaved into two polypeptides that rejoin through their hetero-dimerization (HD) domains at the cell’s surface during expression. The extracellular fragment is composed of multiple epidermal growth factor (EGF)-like repeats that mediate binding to NOTCH ligands; negative regulatory regions (LNRs) that prevent spontaneous activation of unbound NOTCH receptors; and a portion of the HD domain that binds the intracellular NOTCH fragment. Intracellular NOTCH fragments contain a small portion of the HD domain, the transmembrane domain (TM), the recombination signal binding protein (RBP)-Jκ-associated molecule (RAM) domain, ankyrin repeats (ANK), a transcription activation domain (TAD), and a C-terminal proline-, glutamate-, serine-, and threonine-rich (PEST) sequence. The PEST domain interacts with FBXW7 to degrade activated intracellular NOTCH. Upon binding ligand (Jagged 1/2 (JAG1/2) or Delta-like ligand 1/3/4 (DLL1/3/4)) that is expressed on the surface of a neighboring cell, NOTCH receptors are activated in a juxtacrine manner to undergo conformational changes. These changes expose a cleavage site in the TM domain to a disintegrin and metalloproteinase (ADAM) proteases and finally to a γ-secretase complex. Intracellular NOTCH is liberated as activated intracellular NOTCH1 domain (ICN) that migrates to the nucleus and interacts with DNA binding proteins to regulate gene transcription of target genes. DSL, Delta/Serrate/Lag-2; VWC, von Willebrand factor type C domain; PDZ, PSD-95/Dlg/ZO-1.

**Figure 2 cells-09-02677-f002:**
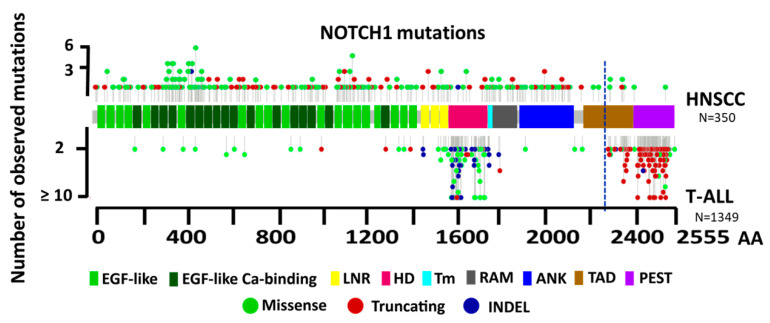
Schematic representation of *NOTCH1* mutation distributions in HNSCC and T cell acute lymphoblastic leukemia (T-ALL). *NOTCH1* mutations from unique patients reported from genomic studies in the COSMIC database and The Cancer Genome Atlas (TCGA) Pan-cancer cohorts were filtered to remove variants that were not predicted to be impactful by either Protein Variation Effect Analyzer (PROVEAN) or Sorting Intolerant From Tolerant (SIFT) scores. All indels and missense alterations predicted to be impactful plus truncating (i.e., frame shift or nonsense) mutations for HNSCC (n = 350) are illustrated on the top half of the plot and those from T-ALL (n = 1349) are on the bottom half of the diagram. A blue dotted line represents the approximate boundary separating the majority of truncating mutations in HNSCC from those found in T-ALL. AA, Amino acids; EGF, Epidermal growth factor; LNR, Negative regulatory region; HD, Hetero-dimerization domain; TM, Transmembrane domain; RAM, RBP-Jκ-associated molecule; ANK, Ankyrin repeats; TAD, Transcription activation domain; PEST, C-terminal proline-, glutamate-, serine-, and threonine-rich sequence; INDEL, Insertion and deletion.

**Figure 3 cells-09-02677-f003:**
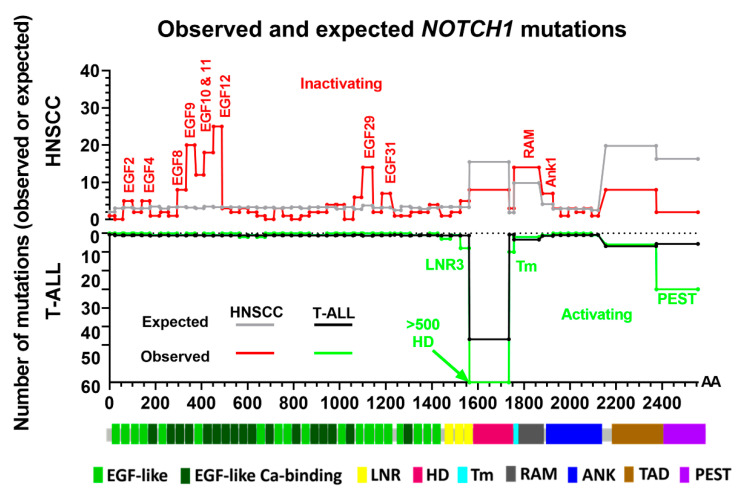
Domains enriched for *NOTCH1* missense and in-frame mutations. Unique *NOTCH1* missense and INDEL mutations (excluding truncations) from T-ALL and HNSCC within the COSMIC database and from the TCGA were filtered to include only those predicted to be impactful by PROVEAN or SIFT scores to obtain the total number of observed impactful mutations for either HNSCC (*n* = 230) or T-ALL (*n* = 843) used in calculations and the plot. The expected number of mutations for each domain was calculated by multiplying the probability for a mutation to occur in a domain (i.e., size of domain/total size of NOTCH1 protein) times the total number of impactful mutations. For T-ALL, the expected and observed mutations were calculated independently for the HD domain and domains outside this region to avoid skewing results.

**Figure 4 cells-09-02677-f004:**
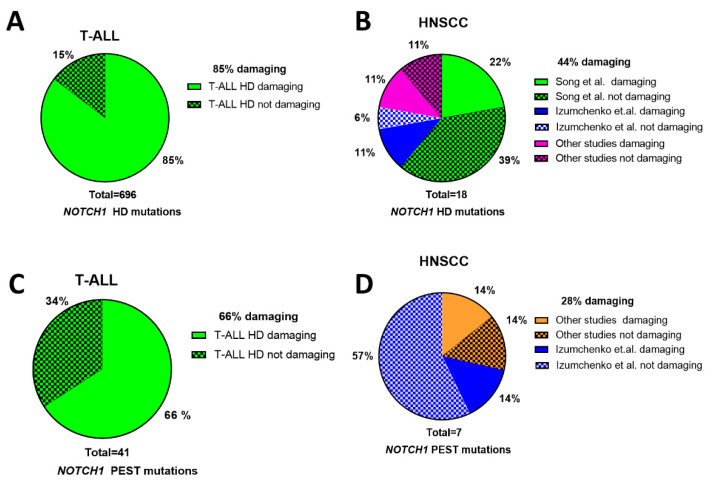
Predicted impact of *NOTCH1* missense mutations from the HD and PEST domains. *NOTCH1* missense mutations occurring in the HD domain for T-ALL (**A**) or HNSCC (**B**) with proportions predicted to be impactful by PROVEAN or SIFT (solid pattern) or tolerated (checkered pattern). *NOTCH1* missense mutations occurring in the PEST domain for T-ALL (**C**) or HNSCC (**D**) with proportions predicted to be impactful by PROVEAN or SIFT (solid pattern) or tolerated (checkered pattern).

**Figure 5 cells-09-02677-f005:**
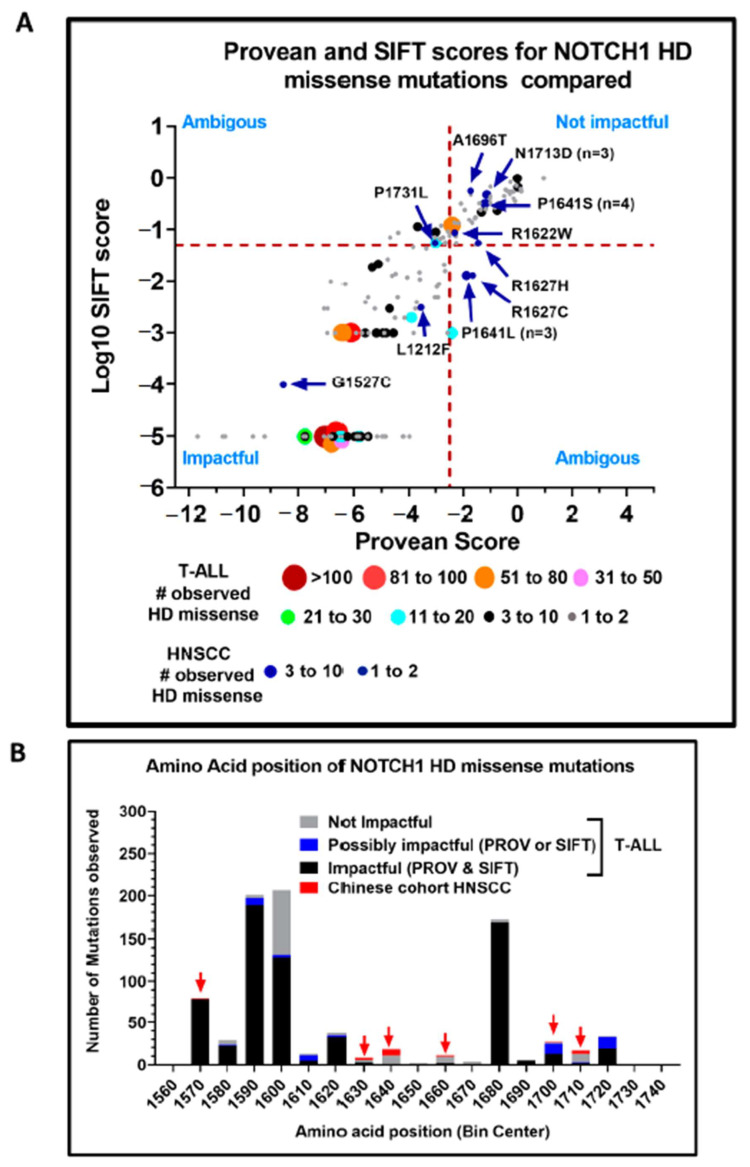
Comparison of potentially activating *NOTCH1* mutations reported from the HD domain from two Chinese HNSCC cohorts [16,17] and T-ALL from the COSMIC database. (**A**) The PROVEAN and SIFT scores for each mutation. For T-ALL, the total numbers of mutations in each region of the graph are indicated by both color and circle size, with increasing diameters signifying increasing numbers of mutations. Mutations observed in HNSCC studies are indicated with blue circles. Mutations in the lower left quadrant with the lowest PROVEAN and SIFT scores are predicted to be the most impactful, while mutations in the upper right quadrant are predicted to be the least impactful. (**B**) Distribution of *NOTCH1* HD domain mutations. Colored bars indicate whether T-ALL mutations are predicted to be impactful by both PROVEAN and SIFT, possibly impactful (i.e., PROVEAN or SIFT), or not impactful by both methods. For comparison, HNSCC *NOTCH1* mutations are represented by the red boxes and bins where HNSCC mutations occur are also indicated with red arrows.

**Figure 6 cells-09-02677-f006:**
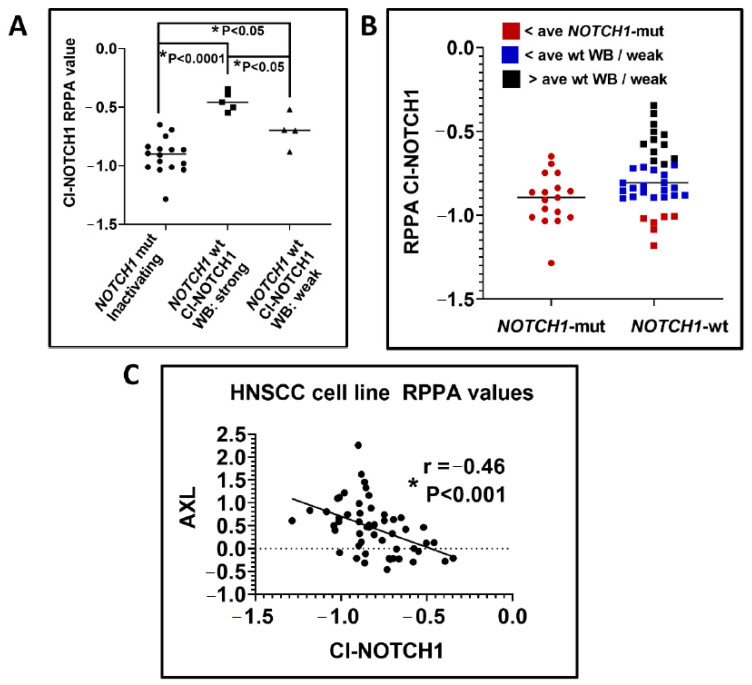
Levels of activated NOTCH1 by reverse phase protein arrays (RPPA) correlate with *NOTCH1* mutational status, immunoblot data, and AXL expression. (**A**) RPPA protein expression of activated cleaved Cl-NOTCH1 (ICN1) is significantly lower in HNSCC cell lines with inactivating *NOTCH1* mutations compared to *NOTCH1*-wt cell lines. Cell lines with strongest Cl-NOTCH1 signal on Western blots (squares) had higher levels of Cl-NOTCH1 by RPPA as well. Significance was determined by a one-way ANOVA followed by Tukey’s multiple comparison test. (**B**) The distribution of Cl-NOTCH1 levels measured by RPPA among *NOTCH1*-wt HNSCC cell lines compared to expression in *NOTCH1*-mut tumors. *NOTCH1*-wt cell lines (red square symbols) with Cl-NOTCH1 RPPA values below the average observed for *NOTCH1* mutants or below the average value for *NOTCH1*-wt cell lines with only weak expression on Western blot (blue square symbols) identify cell lines lacking strong endogenous NOTCH1 pathway activation. Roughly one third of *NOTCH1*-wt cell lines (black square symbols) analyzed had RPPA values in a range that matched cells with strong pathway activation confirmed by immunoblots. (**C**) Strong anti-correlation between Cl-NOTCH1 expression and total AXL protein as measured by RPPA in HNSCC cell lines. The significance of the correlation coefficient was determined with a *t*-test. WB, Western blot; ave, Average; mut, Mutant; wt, Wild type.

**Figure 7 cells-09-02677-f007:**
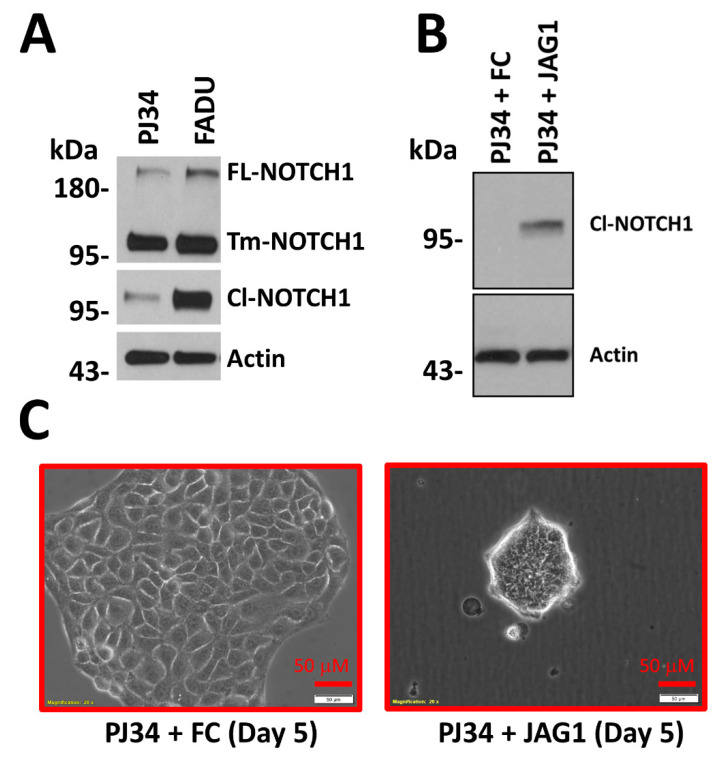
NOTCH1 pathway activation is associated with morphological change in a HNSCC cell line harboring wt *NOTCH1.* (**A**) Strong total NOTCH1 protein expression is evident in two HNSCC cell lines (PJ34 and FaDu) with wt *NOTCH1*, but baseline pathway activation detected with an antibody to ICN1 is much stronger in FaDu cells. (**B**) Cultivation of PJ34 on immobilized NOTCH1 ligand (JAG1) for 16 h dramatically increases the amount of detectable ICN1 compared to growth on immobilized control (FC) protein. (**C**) Morphology changes in PJ34 after growth on JAG1 ligand detectable as early as within 3 days become prominent by day 5, and are characterized by drastic reduction in individual cell size and a shift to form loosely attached spheroids. FL, Full-length; Tm, Transmembrane domain; Cl, Cleaved.

**Figure 8 cells-09-02677-f008:**
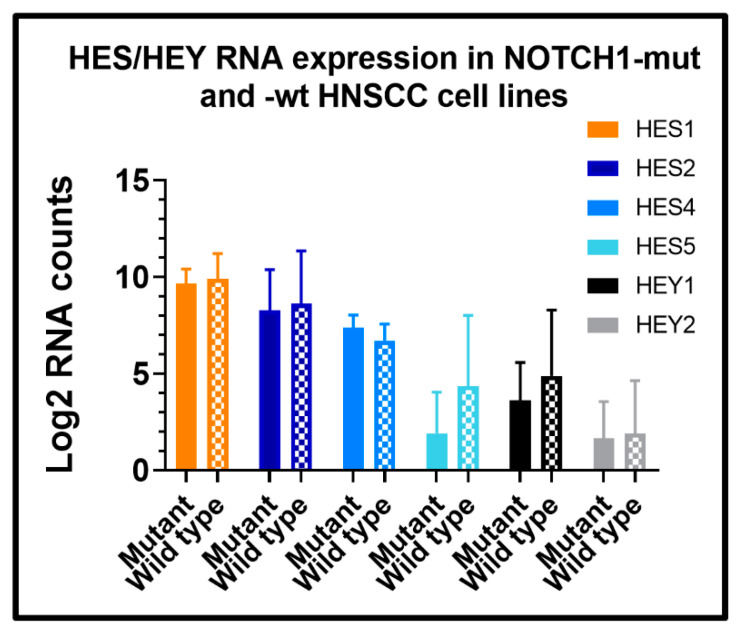
Expression of HES/HEY downstream targets is a poor surrogate for NOTCH1 pathway status. RNA expression of *HES/HEY* family members from five HNSCC cell lines with confirmed high levels of baseline ICN1 were compared to expression levels in six HNSCC cell lines with confirmed homozygous inactivating *NOTCH1* mutations. Solid bars are mutant *NOTCH1* averages ± standard deviation (sd); checkered bars are wt *NOTCH1* averages ± sd. Levels of RNA were detected on an Affymetrix microarray in a large panel of HNSCC cell lines and counts subjected to upper quartile normalization before analysis. None of the differences in *HES/HEY* RNA expression was significantly different, but there was a trend towards increased *HES5* and *HEY* in the *NOTCH1*-wt cells (Table 2). *HES3* RNA was not detected in any cells.

**Table 1 cells-09-02677-t001:** Summary of *NOTCH1* genomic studies.

				HPV-Negative HNSCC				OPSCC/HPV-Positive HNSCC				
Year	Study	Subsite	Cohort Characteristics	#Patients	%Truncating	%Missense	%Total	#Patients	%Truncating	%Missense	%Total	Clinical Associations with *NOTCH1* Mutations
2011	[10]	Mixed	USA	90	10%	9%	19%	30	1%	13%	14%	N.D.
2011	[11]	Mixed	USA	60	7%	7%	14%	14	0%	7%	7%	N.D.
2013	[38]	Gingivo-buccal	South Asian (mixed HPV neg and pos)	50	8%	8%	16%	N.D.	N.D.	N.D.	N.D.	N.D.
2014	[16]	OCSCC	Chinese	51	10%	37%	43%	N.D.	N.D.	N.D.	N.D.	*NOTCH1* mutations significantly associated with worse OS and DFS, and LN mets
2014	[37]	OCSCC	Japanese OCSCC; limited exons sequenced	84	0%	10%	10%	N.D.	N.D.	N.D.	N.D.	*NOTCH1* mutations associated with better DFS, but no difference in OS
2015	[40]	Mixed	USA	69	4%	12%	16%	51%	4%	4%	8%	N.D.
2015	[42]	Mixed	Chinese	50	6%	48%	54%	N.D.	N.D.	N.D.	N.D.	N.D.
2015	[13]	Mixed	USA	246	9%	12%	20%	20	0%	9%	9%	No association found
2016	[36]	Mixed	Chinese	128	5%	12%	22%	N.D.	N.D.	N.D.	N.D.	*NOTCH1* mutations significantly associated with reduced OS and increased recurrence
2017	[41]	Mixed	Recurrent HNSCC; USA	30	10%	10%	20%	21	0%	10%	10%	N.D. for HNSCC
2018	[39]	Mixed	European and Latino	165	7%	16%	23%	15	0	2	13%	N.D.
2018	[43]	Mixed	USA	445	8%	10%	18%	51	0%	6%	6%	N.D.

USA, United States of America; HPV, Human papilloma virus; OS, Overall survival; DFS, Disease-free survival; LN mets, Lymph node metastasis; HNSCC, Head and neck squamous cell carcinoma; OCSCC, Oral cavity squamous cell carcinoma; N.D., Not done; neg, Negative; pos, Positive.

**Table 2 cells-09-02677-t002:** *NOTCH1* mutations in HNSCC cell lines.

HNSCC Cell Line	HPV Status	Protein Change [52,54]	Zygosity	Variant Type	Variant Class	Domain	PROVEAN Score	SIFTA Score	Consensus	Domain	Predicted Function	Experimental Evidence [12,55,56]
MDA1686	Neg	H2018FS	Heterozygous	Frame shift del	Truncating	ANK4	N/A	N/A	N/A	ANK4	Inactivating	WB: NOTCH1 null
UMSCC25	Neg	V489FS	Homozygous	Frame shift del	Truncating	Ca binding EGF_12	N/A	N/A	N/A	Ca binding EGF_12	Inactivating	
HN4	Neg	C344FS	Homozygous	Frame shift ins	Truncating	Ca binding EGF_9	N/A	N/A	N/A	Ca binding EGF_9	Inactivating	WB: NOTCH1 null
PCI15A	Neg	Q1957 *	Heterozygous	Nonsense mutation	Truncating	ANK2	N/A	N/A	N/A	ANK2	Inactivating	WB: very weak NOTCH1
PCI15B	Neg	Q1957 *	Homozygous	Nonsense mutation	Truncating	ANK2	N/A	N/A	N/A	ANK2	Inactivating	WB: NOTCH1 null
MDA1686	Neg	E2008 *	Heterozygous	Nonsense mutation	Truncating	ANK4	N/A	N/A	N/A	ANK4	Inactivating	WB: NOTCH1 null
UMSCC85	Neg	E694 *	Homozygous	Nonsense mutation	Truncating	Ca binding EGF_18	N/A	N/A	N/A	Ca binding EGF_18	Inactivating	
UMSCC47	HPV16	G192 *	Homozygous	Nonsense mutation	Truncating	Ca binding EGF_5	N/A	N/A	N/A	Ca binding EGF_5	Inactivating	WB: NOTCH1 null
UMSCC22A	Neg	E1679 *	Homozygous	Nonsense mutation	Truncating	HD	N/A	N/A	N/A	HD	Inactivating	WB: NOTCH1 null
UMSCC22B	Neg	E1679 *	Homozygous	Nonsense mutation	Truncating	HD	N/A	N/A	N/A	HD	Inactivating	
UMSCC25	Neg	E488A	Homozygous	Missense mutation	SNV	Ca binding EGF_12	Deleterious/ −4.65	Damaging/0.004	Impactful	Ca binding EGF_12	Inactivating	
HN30	Neg	C478F	Homozygous	Missense mutation	SNV	Ca binding EGF_12	Deleterious/ −9.71	Damaging/0.0	Impactful	Ca binding EGF_12	Inactivating	
HN31	Neg	C478F	Homozygous	Missense mutation	SNV	Ca binding EGF_12	Deleterious/ −9.71	Damaging/0.0	Impactful	Ca binding EGF_12	Inactivating	ICN1 absent +/− ligand exp
SCC45	HPV33	G72R	Homozygous	Missense mutation	SNV	EGF_2	Deleterious/ −6.81	Damaging/0.0	Impactful	EGF_2	Inactivating	
MSK922	Neg	C1536Y	Homozygous	Missense mutation	SNV	LNR3	Deleterious/ −10.65	Damaging/0.0	Impactful	LNR3	Weakly activating	
PCI13	Neg	G1753W	Homozygous	Missense mutation	SNV	TM	Deleterious/ −5.83	Damaging/0.01	Impactful	TM	Inactivating	WB: very weak NOTCH1
TR146	Neg	A1524V	Heterozygous	Missense mutation	SNV	LNR3	Neutral/ −0.55	Tolerated/ 0.319	Not Impactful	LNR3	Impact unlikely	
JHU029	Neg	L418del	Homozygous	In frame del	INDEL	Ca binding EGF_11	N/A	N/A	N/A	Ca binding EGF_11	Inactivating	
1483	Neg	F357del	Homozygous	In frame del	INDEL	Ca binding EGF_9	N/A	N/A	N/A	Ca binding EGF_9	Inactivating	

HPV, Human papilloma virus; Neg, Negative; N/A, Not applicable; WB, Western blot; ICN, Intracellular NOTCH; del, Deletion; ins, Insertion; SNV, Single nucleotide variant; INDEL, Insertion and deletion mutation; EGF, Epidermal growth factor-like; ANK, Ankyrin repeats; HD, Hetero-dimerization; LNR3, Lin-12/NOTCH repeats 3; TM, Transmembrane; ICN1, Intracellular NOTCH1.

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
