# Peer review of "NOTCH1 Signaling in Head and Neck Squamous Cell Carcinoma"

_cells, 2020, doi:10.3390/cells9122677_

Round 1
Reviewer 1 Report
Comments
This is a well written, fairly comprehensive, and timely review of the literature pertaining to the role and significance of Notch signaling in HNSCC. My principal criticism is that the authors repeated cite their own unpublished data, which seems inappropriate for a review. It would be better if they simply criticize the data with which their own conflicts, knowing that these are unsettled areas of investigation.
Other criticisms are fairly minor, as follows:
1) The two Notch subunits do not “reassociate” during maturation as stated on page 2; folding occurs independent of cleavage by furin, which occurs in an unstructured region. In fact, furin cleavage is not necessary for Notch function in flies and possibly in mammalian cells.
2) There is no evidence that ICN interacts with “numerous” transcription factor complexes (page 2); strong evidence exists only for RBPJ/MAML complexes.
3) There is no clear structural or functional evidence for a PEST region in Notch4. Figure 1 needs to be modified to reflect this difference across Notch receptors.
4) Page 4; for completeness it would be worth citing Proweller et al (Cancer Res. 2006 Aug 1;66(15):7438-44. doi: 10.1158/0008-5472.CAN-06-0793), which showed that a pan-Notch inhibitor was a potent inducer of cutaneous squamous cell carcinoma.
5) The effects of Notch on transcription are very rapid (30 minutes or so); thus, 5 days of culture on ligand will likely induce many secondary effects. This does not negate the possible significance of changes in gene expression at this time point, but it should be acknowledged that these are likely to be secondary to whatever the direct targets of Notch are in this cell context. The identity of immediate downstream effects of Notch in squamous cells remains unsettled, though some new data have recently appeared from Loganathan et al (see below).
6) The term “high basal ICN1” is a bit nebulous. Notch1 activation is a normal facet of squamous differentiation in suprabasal transit amplifying cells. Are lines with “high basal ICN1” merely undergoing some level of differentiation in culture?
7) It is true that basal cells are ICN1 low and suprabasal cells are ICN1 high; but is there evidence (as suggested) that it is activation of Notch1 in basal cells that somehow directs these asymmetric divisions? Or is movement of these cells to a suprabasal position what then leads to Notch activation?
8) The authors should cite and discuss in brief the recent paper of Loganathan et al., which describes interaction of Notch LOF with a number of other signaling pathways, including the PI3K pathway.
Author Response
This is a well-written, fairly comprehensive, and timely review of the literature pertaining to the role and significance of Notch signaling in HNSCC. My principal criticism is that the authors repeated cite their own unpublished data, which seems inappropriate for a review. It would be better if they simply criticize the data with which their own conflicts, knowing that these are unsettled areas of investigation.
We appreciate this comment and have removed many references to unpublished data. We now include three new figures (Figs. 6, 7, and 8) and an additional supplementary table with primary data from our own experiments to support some alternative interpretations for active areas of investigation. We have also substantially reworded some of the text to make it less critical of existing publications but point out limitations and possible alternative explanations.
Other criticisms are fairly minor, as follows:
1) The two Notch subunits do not “reassociate” during maturation as stated on page 2; folding occurs independent of cleavage by furin, which occurs in an unstructured region. In fact, furin cleavage is not necessary for Notch function in flies and possibly in mammalian cells.
We have now revised the text as follows to more accurately detail NOTCH1 maturation: “NOTCH1 is first synthesized as a precursor protein that folds and initially undergoes cleavage to produce extracellular and intracellular/ transmembrane-associated peptides that remain tightly bound through their hetero-dimerization domains (HD) during transport to the cell plasma membrane.”
2) There is no evidence that ICN interacts with “numerous” transcription factor complexes (page 2); strong evidence exists only for RBPJ/MAML complexes.
We altered the text to more clearly explain interaction with transcriptional coactivators and repressors as follows: “Cleavage occurs within the transmembrane (TM) region, releasing the intracellular NOTCH1 domain (ICN1), sometimes referred to as NOTCH intracellular domain (NICD) or cleaved (cl) NOTCH1, which migrates to the nucleus where it regulates transcription of many genes.
In the nucleus, ICN1 forms a ternary complex with recombination signal binding protein Jk [RBP-Jk, aslo called Epstein-Barr virus (EBV) C promoter binding factor 1 (CBF1)/Suppressor of Hairless (Su (H))/Lag-1 (CSL)] and mastermind like transcriptional coactivator 1 (MAML1). The ICN1/RBP-Jk/MAML1 complex activates transcription of canonical downstream NOTCH1 targets from the Hairy and enhancer of split-1 (HES) and Hairy/enhancer-of-split related with YRPW motif protein 1 (HEY) gene families, which are themselves transcriptional repressors. Additionally, activated ICN1/RBP-Jk/MAML1 recruits multiple other coactivators or repressors of gene expression, including histone modifiers.”
3) There is no clear structural or functional evidence for a PEST region in Notch4. Figure 1 needs to be modified to reflect this difference across Notch receptors.
Yes, that was an oversight when making the figure, which has now been corrected. Thank you.
4) Page 4; for completeness it would be worth citing Proweller et al (Cancer Res. 2006 Aug 1;66(15):7438-44. doi: 10.1158/0008-5472.CAN-06-0793), which showed that a pan-Notch inhibitor was a potent inducer of cutaneous squamous cell carcinoma.
We added this reference.
5) The effects of Notch on transcription are very rapid (30 minutes or so); thus, 5 days of culture on ligand will likely induce many secondary effects. This does not negate the possible significance of changes in gene expression at this time point, but it should be acknowledged that these are likely to be secondary to whatever the direct targets of Notch are in this cell context. The identity of immediate downstream effects of Notch in squamous cells remains unsettled, though some new data have recently appeared from Loganathan et al (see below).
Yes, we have a manuscript in preparation profiling much earlier signaling events. The purpose of looking at later time point changes is to understand the phenotypic consequences of NOTCH1 signaling that become quite apparent from 3 days to 5 days after growth in the presence of JAG1. We now include primary data showing morphological changes that accompany growth on JAG1 (New Figure 7), which likely arise from activation of genes involved in early differentiation (now Supplementary Table 3).
6) The term “high basal ICN1” is a bit nebulous. Notch1 activation is a normal facet of squamous differentiation in suprabasal transit amplifying cells. Are lines with “high basal ICN1” merely undergoing some level of differentiation in culture?
Exactly, we hypothesize that NOTCH1 activation in HNSCC tumor lines turns on a gene expression program that to some degree recapitulates early differentiation (New Supplemental Table 3, and unpublished data from our manuscript in preparation or presented previously at meetings). We now include a Western blot (New Figure 7) that clarifies what we mean by high basal level of ICN1. In Figure 7, we show that PJ43 has relatively weak ICN1 expression compared to FADU cells, considering their total NOTCH1 expression is not that different. If PJ34 is exposed to NOTCH1 ligand, then levels of ICN1 go up at least 5 to 10-fold and are more consistent with levels seen in cell lines like FADU in the absence of externally added ligand. We know these cell lines express NOTCH1 ligands and theorize that there are differences in ligand level and cis-inhibition. We have revised text in Section 4.1 (HNSCC cell line models for studying NOTCH1) and Section 4.3 (NOTCH1 pathway in HNSCCs harboring wt NOTCH1) and to include Western blots and explain how they correlate with varying levels of endogenous expression of ICN1 detected by RPPA analysis in HNSCC wt tumor lines (New Supplementary Table 2).
7) It is true that basal cells are ICN1 low and suprabasal cells are ICN1 high; but is there evidence (as suggested) that it is activation of Notch1 in basal cells that somehow directs these asymmetric divisions? Or is movement of these cells to a suprabasal position what then leads to Notch activation?
Excellent question: “chicken or the egg?” Evidence from the literature suggests that ICN1 signaling is the trigger for asymmetric divisions leading to early differentiation and loss of integrin expression associated with detachment from the basement membrane. We now include some additional text in the review that clarifies this point and cites the work by Rangarajan et al. [72], which showed conditional knockout of NOTCH1 in mouse skin led to expansion of cells expressing the basal markers K14, ITGB1, and ITGB4. Rangarajan et al. further showed that activation of NOTCH1 in keratinocytes led to reduced integrin expression. Lineage tracing experiments with a dominant negative MAML1-GFP fusion protein demonstrate that loss of NOTCH signaling leads to increased progenitor proliferation and decreased rate of differentiating cell stratification [71].
8) The authors should cite and discuss in brief the recent paper of Loganathan et al., which describes interaction of Notch LOF with a number of other signaling pathways, including the PI3K pathway.
We added this study in section 4.3
Reviewer 2 Report
In this review ”NOTCH1 Signaling in Head and Neck Squamous Cell Carcinoma ” by Shah P.A. et al. the authors describe the role of NOTCH1 signaling in Head and Neck Squamous Cell Carcinoma (HNSCC). On the whole the review covers comprehensively the topic proposed i.e. NOTCH1 signaling in HNSCC, however major revisions with significant English editing needs to be done to render the manuscript more appealing and smooth to read.
Major concerns:
It is clear that the authors have made an effort to document all information concerning NOTCH1 in HNSCC, however in its present form the manuscript is rather difficult to read with some sections seemingly redundant and long-winded. In its present form too much emphasis is put on the results of abstracts presented by the group (reference 51/52) and not published (with peer review) or prolonged summaries of their published (reference 12)/or unpublished results. For example all of section 4.3 “NOTCH1 pathway activation in HNSCC harboring wt NOTCH1” is based on an Abstract (reference 51). The authors should put some of their original data to justify the comments made or down-tune the importance of their unpublished results.
Summarizing, numerous sections are too long (redundant) and need to be shortened/summarized or merged. For example paragraph 3.4 and 4.5 could be merged under one heading. Paragraphs 4.1 to 4.4 describe their published data and could be merged in a single paragraph. A cartoon that shows the main putative onco-suppressive and oncogenic roles of NOTCH1 in HNSCC would be useful.
A very recent study “Rare driver mutations in head and neck squamous cell carcinomas converge on NOTCH signaling” by S.K. Loganathan et al. Science 2020 should also be commented to render the review more up-to-date and inserted amongst the references. In addition, the authors could think about adding a paragraph on NOTCH directed therapies.
Minor concerns: English and text formatting/editing needs to be done on the manuscript including:
-the authors should check carefully their manuscript and make sure ALL abbreviations used have been initially written in full e.g. NOTCH1 (page 2), ADAM, MAML1, HES, HEY etc. Also, note that the abstract, main text and figure/table/scheme captions are treated separately for abbreviations, as mentioned in the guide for authors.
-Authors should check the correct abbreviations for genes/ proteins/domains (e.g. CTNNAL1 instead of CTTNL1…)
-Gene names require italics. As mentioned in the guide for authors, latin terms do not need to be italicized, but should be consistent throughout the text (in vitro/vivo; e.g; i.e).
- The concluding sentence/phrase of the “Introduction” starting with “Given the likely importance… harbor NOTCH1 mutations” is too long and needs editing to make it sharper and more concise.
-On page 2, Section 2 the HD (hybridization domain) should be hetero-dimerization domain. It would be better to use the term HD domain, rather than just HD. This also applies to Figure 1 legend where again the term hybridization domain is incorrectly used.
- Paragraph 3.3: The comparison of the distribution of NOTCH1 mutations between HSNCC and T-ALL is introduced rather abruptly. The authors should better contextualize the importance of Notch signaling in T-ALL before and also explain the reasons for such a comparison.
-Figure 2 legend needs to be edited to make the concepts clearer e.g. “Upon binding ligand JAG1/2 or DLL3/4……
-juxtracrine should be juxtacrine
-the sentence “Certain leukemias arise with inactivating FBXW7……” should be rephrased as the current sentence is misleading.
- All tables need to be edited as most headings are cut..e.g. Table 1. Stud y Subsit e etc. (although this may be a formatting problem). In addition, a reference column should be included in Table 2.
-On page 6, the sentence “Interestingly…. equivalently …..to the same degree…..” are a repetition, one should be eliminated.
-On page 6, the term “reduced burden” is inappropriate.
-Always on page 6, paragraph “3.3 Structural Characterization of NOTCH1 mutations” there is an inconsistency in the numbers i.e. the text mentions 350 and 1349 potentially impactful mutations (and on Fig.2 cartoon) while the Figure 2 legend states 334 and 1339 impactful mutations. Which is correct?
-On page 6, the last two sentences are not clear to the reviewer. The authors need to edit the sentences to make them clearer.
-In Figures 2 and 3, the axes need to be labeled. Authors should check sample size (N) throughout their Figure legends and text to render them consistent.
-The significance of the finding “impactful” as defined by the authors is debatable, as suggesting that 15% (HD mutations) and 24% (PEST domain) of NOTCH1 mutations in T-ALL are not damaging does not seem appropriate.
-On page 10, it is stated that high basal expression of ICN1 is observed for some HNSCC cell lines wt NOTCH1. This is very surprising. The reference is one of their Abstracts. Are the authors confident about the antibody specificity? How would NOTCH1 activation occur in these cell lines? Also, the presence of total NOTCH1 protein does not necessarily correlate with activated NOTCH1 (ICN1), if the pathway is not activated.
-From section 4.2 onward page numbering is lost and not consecutive with the previous sections. Also, the title has a typo i.e. altlers instead of alters.
-Figure 4. The figure is somehow confusing and redundant. Pie charts could be excluded and mutation frequencies in T-ALL vs HNSCC (i.e. 85% vs 44%; 66% vs 28%) could be simply added as percentages to Figure 5, without the need of an additional Figure.
-Figure 5 legend needs English editing for the first two sentences. Also for Figure 5a: the captions says that HNSCC mutations are indicated by green circles, but circles are blue in the figure.
- The heading of section 6 needs editing, and the last sentence “However,….NOTCH pathway through…” needs to be edited.
-the authors should explain better what they mean when they say that in HNSCC, NOTCH1 mutations mainly occur before the TAD domain and “ are likely LOF because not enough NOTCH1 protein is made to function properly”.
-Always in the “conclusion” section, the sentence “NOTCH1 missense mutations….where NOTCH1 mutations are activating and there were are more mutations in the HD” needs to be edited . Also the sentence after that could be edited to make the point clearer. Further, the authors mention a 93 gene NOTCH1 signature. Is this a published signature? The authors should specify better how this signature was obtained.
Author Response
In this review ”NOTCH1 Signaling in Head and Neck Squamous Cell Carcinoma ” by Shah P.A. et al. the authors describe the role of NOTCH1 signaling in Head and Neck Squamous Cell Carcinoma (HNSCC). On the whole the review covers comprehensively the topic proposed i.e. NOTCH1 signaling in HNSCC, however major revisions with significant English editing needs to be done to render the manuscript more appealing and smooth to read.
We edited the document to improve readability.
Major concerns:
It is clear that the authors have made an effort to document all information concerning NOTCH1 in HNSCC, however in its present form the manuscript is rather difficult to read with some sections seemingly redundant and long-winded. In its present form too much emphasis is put on the results of abstracts presented by the group (reference 51/52) and not published (with peer review) or prolonged summaries of their published (reference 12)/or unpublished results. For example all of section 4.3 “NOTCH1 pathway activation in HNSCC harboring wt NOTCH1” is based on an Abstract (reference 51). The authors should put some of their original data to justify the comments made or down-tune the importance of their unpublished results.
Great suggestions. We have now added some of our primary data in Figures 6, 7, 8, and Supplementary Tables 2 and 3. We also revised the text to remove emphasis on unpublished data and instead explain limitations in our current understanding of published studies or present alternative interpretations for consideration.
Summarizing, numerous sections are too long (redundant) and need to be shortened/summarized or merged. For example paragraph 3.4 and 4.5 could be merged under one heading. Paragraphs 4.1 to 4.4 describe their published data and could be merged in a single paragraph. A cartoon that shows the main putative onco-suppressive and oncogenic roles of NOTCH1 in HNSCC would be useful.
We have edited the manuscript to reduce redundancy. Sections 3.4 and 4.5 address distinct topics – NOTCH1 mutations and NOTCH signaling respectively and cannot be combined. The overwhelming evidence is for a tumor suppressive role for NOTCH signaling in HNSCC so we did not create a figure demonstrating the suppressive and oncogenic roles of NOTCH1 in HNSCC.
A very recent study “Rare driver mutations in head and neck squamous cell carcinomas converge on NOTCH signaling” by S.K. Loganathan et al. Science 2020 should also be commented to render the review more up-to-date and inserted amongst the references. In addition, the authors could think about adding a paragraph on NOTCH directed therapies.
We added the Loganathan study in section 4.3
We added information about Notch-directed therapy to section 7 that addresses therapies targeting NOTCH1 mutant HNSCC.
Minor concerns: English and text formatting/editing needs to be done on the manuscript including:
-the authors should check carefully their manuscript and make sure ALL abbreviations used have been initially written in full e.g. NOTCH1 (page 2), ADAM, MAML1, HES, HEY etc. Also, note that the abstract, main text and figure/table/scheme captions are treated separately for abbreviations, as mentioned in the guide for authors.
We corrected these errors.
-Authors should check the correct abbreviations for genes/ proteins/domains (e.g. CTNNAL1 instead of CTTNL1…)
We corrected these errors.
-Gene names require italics. As mentioned in the guide for authors, Latin terms do not need to be italicized, but should be consistent throughout the text (in vitro/vivo; e.g; i.e).
We corrected these errors.
- The concluding sentence/phrase of the “Introduction” starting with “Given the likely importance… harbor NOTCH1 mutations” is too long and needs editing to make it sharper and more concise.
We revised this sentence.
-On page 2, Section 2 the HD (hybridization domain) should be hetero-dimerization domain. It would be better to use the term HD domain, rather than just HD. This also applies to Figure 1 legend where again the term hybridization domain is incorrectly used.
We corrected these errors.
- Paragraph 3.3: The comparison of the distribution of NOTCH1 mutations between HSNCC and T-ALL is introduced rather abruptly. The authors should better contextualize the importance of Notch signaling in T-ALL before and also explain the reasons for such a comparison.
We explained the reason for the comparison.
-Figure 2 legend needs to be edited to make the concepts clearer e.g. “Upon binding ligand JAG1/2 or DLL3/4……
We edited the legend for Figure 1.
-juxtracrine should be juxtacrine
We corrected this error.
-the sentence “Certain leukemias arise with inactivating FBXW7……” should be rephrased as the current sentence is misleading.
We edited this sentence.
- All tables need to be edited as most headings are cut..e.g. Table 1. Stud y Subsit e etc. (although this may be a formatting problem). In addition, a reference column should be included in Table 2.
The tables are formatted correctly in the submitted version so the cut headings are likely a conversion issue. We added references to Table 2.
-On page 6, the sentence “Interestingly…. equivalently …..to the same degree…..” are a repetition, one should be eliminated.
We corrected this error.
-On page 6, the term “reduced burden” is inappropriate.
We corrected this error.
-Always on page 6, paragraph “3.3 Structural Characterization of NOTCH1 mutations” there is an inconsistency in the numbers i.e. the text mentions 350 and 1349 potentially impactful mutations (and on Fig.2 cartoon) while the Figure 2 legend states 334 and 1339 impactful mutations. Which is correct?
We corrected these errors and have now revised both the text and figure legends corresponding to Figures 2 and 3. We apologize for the incorrect information. In Figure 2, all mutations including truncating (i.e., frame shifts and nonsense) as well as missense (predicted to be impactful by PROVEN or SIFT) and INDELs are included to give 350 total for HNSCC and 1349 for T-ALL. In Figure 3, the truncating mutations were excluded for probability calculations, which left 230 for HNSCC and 843 for T-ALL to be considered.
-On page 6, the last two sentences are not clear to the reviewer. The authors need to edit the sentences to make them clearer.
We edited these sentences.
-In Figures 2 and 3, the axes need to be labeled. Authors should check sample size (N) throughout their Figure legends and text to render them consistent.
We added the Y axis legends (e.g., Number of mutations) to Figures 2 and 3, and added the AA (amino acid) symbols on the X axis legends to indicate the numbers are AA positions.
-The significance of the finding “impactful” as defined by the authors is debatable, as suggesting that 15% (HD mutations) and 24% (PEST domain) of NOTCH1 mutations in T-ALL are not damaging does not seem appropriate.
It may seem counter intuitive, but the math absolutely backs up our claim. We have now expanded text in section 3.4 and added a new table (Supplementary Table 1) that explains our calculations and analysis. It is widely accepted that mutations occur in a more or less random fashion throughout genes in tumors, but there is selective outgrowth (i.e., over-representation) only when the mutation actually conveys some advantage. If there is no advantage, then the mutations will occur according to a random chance model. The probability of obtaining non-impactful mutations in the HD and PEST domains by chance can be calculated as explained now in Section 3.4 and are actually quite high (i.e., 70-75%) compared to impactful mutations (See Supplementary Table 1 and below). It is only because there is a positive selection for impactful mutations in these domains in T-ALL that we actually see the observed ratios become reversed. Therefore, it is not unreasonable to see 15% and 34% (now corrected in Figure 4) of mutations predicted to be not impactful as this is about 2 to 4 times lower than what we expect by chance (P<0.0001) using binomial probability.
-On page 10, it is stated that high basal expression of ICN1 is observed for some HNSCC cell lines wt NOTCH1. This is very surprising. The reference is one of their Abstracts. Are the authors confident about the antibody specificity? How would NOTCH1 activation occur in these cell lines? Also, the presence of total NOTCH1 protein does not necessarily correlate with activated NOTCH1 (ICN1), if the pathway is not activated.
We were also surprised by this finding early on but have validated it on many occasions. We are very confident in the specificity of the anti-Cl-NOTCH1 antibody from Cell Signaling and have validated it many times using HN31 cells that express abundant total NOTCH1 but no detectable Cl-NOTCH1 because of an inactivating point mutation in the EGF-like domain. We now include a Western blot in our new Figure 7 showing one example and have added RPPA data and additional figures that indicate roughly how many HNSCC cell lines have high basal ICN1. We believe this occurs because most of the cell lines also express some of the NOTCH ligands, such as JAG1 and JAG2. Very likely it is the ratio between expression of ligands and all the NOTCH receptors that determines how much ICN1 activation occurs verses how much cis-inhibition takes place. Although beyond the scope of this review, we are submitting a manuscript with much more characterization of these NOTCH1 wt cell lines with high basal ICN1. In that paper, we present some experimental evidence that chronic ICN1 leads to compensatory downregulation of downstream targets and may attenuate any growth inhibitory phenotype. Furthermore, HNSCC cell lines with high basal ICN1 do not depend at all on the pathway for growth or survival. More likely, they just tolerate it.
-From section 4.2 onward page numbering is lost and not consecutive with the previous sections. Also, the title has a typo i.e. altlers instead of alters.
We corrected these errors.
-Figure 4. The figure is somehow confusing and redundant. Pie charts could be excluded and mutation frequencies in T-ALL vs HNSCC (i.e. 85% vs 44%; 66% vs 28%) could be simply added as percentages to Figure 5, without the need of an additional Figure.
We apologize if the pie charts in Figure 4 seemed redundant and confusing. There are actually two separate lines of reasoning conveyed in Figure 4 and Figure 5. One way to examine whether mutations are likely drivers is to compare the overall frequency of observed mutations predicted to be impactful verses the expected frequency, which is what is done if Figure 4 for the HD and PEST domains. In this case, the T-ALL pie chart data acts like a positive control. Independent of that, we can look at the physical distribution or localization of impactful and non-impactful mutations within a domain and compare it to a cancer where we know impactful mutations are really doing something. This helps identify exactly which amino acid regions within a domain are important. That is what we are showing in Figure 5, which illustrates that the so called “activating mutations” from the Chinese cohort do not physically overlap with those found in T-ALL.”
-Figure 5 legend needs English editing for the first two sentences. Also for Figure 5a: the captions says that HNSCC mutations are indicated by green circles, but circles are blue in the figure.
We edited the legend and corrected the error.
- The heading of section 6 needs editing, and the last sentence “However,….NOTCH pathway through…” needs to be edited.
We edited the heading.
-the authors should explain better what they mean when they say that in HNSCC, NOTCH1 mutations mainly occur before the TAD domain and “ are likely LOF because not enough NOTCH1 protein is made to function properly”.
We have revised the text to make the explanation more clear as follows: “Because canonical NOTCH1 signaling is enhanced in T-ALL, it can be inferred from the observed distribution of mutations in Figure 2 that truncations happening before the approximate midpoint of the TAD likely weaken or inactivate ICN1 activity because they are practically nonexistent in T-ALL. Consequently, at least a portion of the TAD domain is likely required for full NOTCH1 function, consistent with reports that the TAD domain is required for induction of T-cell leukemia. This is further supported by the distribution of truncations in HNSCC, where the vast majority occur before the TAD and are likely inactivating because they lack a crucial TAD region for activity.”
-Always in the “conclusion” section, the sentence “NOTCH1 missense mutations….where NOTCH1 mutations are activating and there were are more mutations in the HD” needs to be edited. Also the sentence after that could be edited to make the point clearer.
We edited the conclusion section as suggested.
Further, the authors mention a 93-gene NOTCH1 signature. Is this a published signature? The authors should specify better how this signature was obtained.
Our apologies for trying to include too much unpublished data. All text referring to the 93-gene signature and our own inferences about NOTCH1 activation in patient samples using the signature has now been removed. That data will be included in a manuscript now in preparation, where it is more appropriate.
Reviewer 3 Report
This is a well written review on NOTCH1, which analyzes the literature in a meaningful way to distill the information into an easily accessible article. The manuscript identifies current issues in the field as well as some of the limitations of recently published papers that need to be addressed as the field moves forward.
Author Response
This is a well written review on NOTCH1, which analyzes the literature in a meaningful way to distill the information into an easily accessible article. The manuscript identifies current issues in the field as well as some of the limitations of recently published papers that need to be addressed as the field moves forward.
Thank you for your review.
Round 2
Reviewer 1 Report
The authors have satisfactorily addressed most of the prior critiques. They have retained a PEST domain in Figure 1 on Notch4, which is not supported by protein sequence or functional studies. At a minimum, they should somehow indicate that the existence of a function PEST domain in Notch4 is unproven.
Author Response
Figure 1 does not contain a PEST domain (purple) in NOTCH4.
Reviewer 2 Report
In this revised version of the review ”NOTCH1 Signaling in Head and Neck Squamous Cell Carcinoma ” by Shah P.A. et al. the authors describe the role of NOTCH1 signaling in Head and Neck Squamous Cell Carcinoma (HNSCC). The authors have responded constructively to the comments made. The readability (and impact) of the review has been considerably improved.
Minor concerns:
-The last sentence of the introduction is very difficult to read, the authors should consider to abbreviate it in some way, or break it up in more sentences.
-There are some issues with the formatting of the text i.e. on page 3, the sentence …(RAM) domain should be linked to the next line …seven ankyrin (ANK); on page 11, the line..mimic NOTCH1- should be linked to the next line..signaling by..; on page 8 (?), the line..staining scattered- should be linked to the next line..throughout tumor nests..;
-In Figure 6, panel A. It should be NOTCH1 mut and not NOTCH1 mt. Which statistical analysis was done to determine the P value? The legend should also describe the black squares shown in Figure 6B.
-After page 20, the numbering is lost (i.e. the page numbers start again with 1)
- In Figure 7 (panels A and B) please add molecular weights to the blots
-In the paragraph 4.4 (putative page 6) please delete the comma after i.e. (i.e.,)……
-Figure 8 is missing
Author Response
Thank you for your careful review of this manuscript. We appreciate your attention to detail.
-The last sentence of the introduction is very difficult to read, the authors should consider to abbreviate it in some way, or break it up in more sentences.
We revised this one sentence into two, clearer sentences.
-There are some issues with the formatting of the text i.e. on page 3, the sentence …(RAM) domain should be linked to the next line …seven ankyrin (ANK); on page 11, the line..mimic NOTCH1- should be linked to the next line..signaling by..; on page 8 (?), the line..staining scattered- should be linked to the next line..throughout tumor nests..;
We corrected this error on page 8. However, we do not have this formatting issue on pages 3 and 11. We suspect that errors may have been introduced in the conversion of the document for review. We will carefully check for such errors in the publishing proofs.
-In Figure 6, panel A. It should be NOTCH1 mut and not NOTCH1 mt. Which statistical analysis was done to determine the P value? The legend should also describe the black squares shown in Figure 6B.
We corrected these errors.
-After page 20, the numbering is lost (i.e. the page numbers start again with 1)
The numbering is continuous in our version except on the final page that has now been corrected. We suspect that errors may have been introduced in the conversion of the document for review.
- In Figure 7 (panels A and B) please add molecular weights to the blots
We added the molecular weights.
-In the paragraph 4.4 (putative page 6) please delete the comma after i.e. (i.e.,)……
As you know, in American English a comma should follow “i.e.” but in the British style, there is no comma. We used the American style throughout this review. However, if the editor wishes to remove all those commas then she may do so with no objection from us because the change will not affect the substance of our review.
-Figure 8 is missing
We included this figure.